# TrajGPT: Irregular Time-Series Representation Learning for Health Trajectory Analysis

## Abstract

In many domains, such as healthcare, time-series data is often irregularly sampled with varying intervals between observations. This poses challenges for classical time-series models that require equally spaced data. To address this, we propose a novel time-series Transformer called **Trajectory Generative Pre-trained Transformer (TrajGPT)**. TrajGPT employs a novel Selective Recurrent Attention (SRA) mechanism, which utilizes a data-dependent decay to adaptively filter out irrelevant past information based on contexts. By interpreting TrajGPT as discretized ordinary differential equations (ODEs), it effectively captures the underlying continuous dynamics and enables time-specific inference for forecasting arbitrary target timesteps. Experimental results demonstrate that TrajGPT excels in trajectory forecasting, drug usage prediction, and phenotype classification without requiring task-specific fine-tuning. By evolving the learned continuous dynamics, TrajGPT can interpolate and extrapolate disease risk trajectories from partially-observed time series. The visualization of predicted health trajectories shows that TrajGPT forecasts unseen diseases based on the history of clinically relevant phenotypes (i.e., contexts).

## 1 Introduction

Time-series representation learning plays a crucial role in various domains, as it facilitates the extraction of generalizable temporal patterns from large-scale, unlabeled data, which can then be adapted for diverse tasks (Ma et al., 2023). However, a major challenge arises when dealing with irregularly-sampled time series, in which observations occur at uneven intervals (Li & Marlin, 2020). This irregularity poses challenges for classical time-series models that are restricted to regular sampling (Ayala Solares et al., 2020; Zhang et al., 2022). This issue is particularly significant in the healthcare domain, since longitudinal electronic health records (EHRs) are updated sporadically during outpatient visits or inpatient stays (Zhang et al., 2022). Moreover, individual medical histories often span a limited timeframe due to a lack of historical digitization, incomplete insurance coverage, and fragmented healthcare systems (Wornow et al., 2023). These challenges make it difficult for time-series models to capture the underlying trajectory dynamics (Amirahmadi et al., 2023). Addressing these challenges requires the development of novel representation learning techniques that can extract generalizable temporal patterns from irregularly-sampled data through next-token prediction pre-training. The pre-trained model is then applied to forecast trajectories based on the learned transferable patterns, even when patient data is only partially observed.

Recent advances in modeling irregularly-sampled time series have been achieved through specialized deep learning architectures (Che et al., 2018; Horn et al., 2020; Rubanova et al., 2019; Shukla & Marlin, 2021; Zhang et al., 2022). However, these models fall short in pre-training generalizable representations. While time-series Transformer models have gained attention, they are primarily designed for consecutive data and fail to account for irregular intervals between observations (Nie et al., 2023; Zhou et al., 2021; Wu et al., 2021). To handle both regular and irregular time series, TimelyGPT incorporates relative position embedding to capture positional information in varying time gaps (Song et al., 2024a). BiTimelyGPT extends this by pre-training bidirectional representations for discriminative tasks (Song et al., 2024b). Despite these improvements, both models rely on a data-independent decay, which is not content-aware and thus cannot fully capture complex temporal dependencies in healthcare data. The key challenge remains to develop an effective representation learning approach that extracts meaningful patterns from irregularly-sampled data.

In this study, we propose **Trajectory Generative Pre-trained Transformer (TrajGPT)** for irregular time-series representation learning. Our research offers four major contributions: First, it introduces a **Selective Recurrent Attention (SRA)** mechanism with a data-dependent decay, enabling the model to adaptively forget irrelevant past information based on contexts. Second, by interpreting TrajGPT as discretized ODEs, it effectively captures the continuous dynamics in irregularly-sampled data; This enables TrajGPT to perform interpolation and extrapolation in both directions, allowing for a novel time-specific inference for accurate forecasting. Third, TrajGPT demonstrates strong zero-shot performance across multiple tasks, including trajectory forecasting, drug usage prediction, and phenotype classification. Finally, TrajGPT offers interpretable health trajectory analysis, enabling clinicians to align the extrapolated disease progression trajectory with underlying patient conditions.

## 2 RELATED WORKS

### 2.1 TIME-SERIES TRANSFORMER MODELS

Time-series Transformer models have demonstrated strong performance in modeling temporal dependencies through attention mechanisms (Wen et al., 2023). Informer introduces ProbSparse self-attention to extract key information by halving cascading layer input (Zhou et al., 2021). Autoformer utilizes Auto-Correlation to capture series-wise temporal dependencies (Wu et al., 2021). FEDformer adopts Fourier-enhanced attention to capture frequency-domain relationships (Zhou et al., 2022). PatchTST compresses time series into patches and forecasts all timesteps using a linear layer (Nie et al., 2023). Despite their effectiveness, these methods fail to account for irregular time intervals. TimelyGPT and BiTimelyGPT address this limitation by encoding irregular time gaps with relative position embedding (Song et al., 2024a;b). However, these models rely on a data-independent decay, whereas TrajGPT introduces a data-dependent decay to adaptively forget irrelevant information based on contexts. PrimeNet designs a time-sensitive contrastive learning and a masking-and-reconstruction task for irregular time-series representation learning (Chowdhury et al., 2023). ContiFormer integrates ODEs into attention's key and value matrices to model continuous dynamics (Chen et al., 2024). However, it demands significantly more computing resources than a standard Transformer with quadratic complexity, due to the slow process of solving ODEs. In contrast, TrajGPT models continuous dynamics by pre-training on irregularly-sampled data with efficient linear training complexity and constant inference complexity.

### 2.2 ALGORITHMS DESIGNED FOR IRREGULARLY-SAMPLED TIME SERIES

Various techniques have been developed to model irregular temporal dependencies through specialized architectures. GRU-D captures temporal dependencies by applying exponential decay to hidden states (Che et al., 2018). SeFT adopts a set function based approach, where each observation is modeled individually and then pooled together (Horn et al., 2020). RAINDROP captures irregular temporal dependencies by representing data as separate sensor graphs (Zhang et al., 2022). mTAND employs a multi-time attention mechanism to learn irregular temporal dependencies (Shukla & Marlin, 2021). In continuous-time approaches, neural ODEs use neural networks to model complex ODEs, offering promising interpolation and extrapolation solutions (Chen et al., 2018). ODE-RNN further enhances this by updating RNN hidden states with new observations (Rubanova et al., 2019). HeTVAE addresses sparse input with an uncertainty-aware multi-time attention network and represents variable uncertainty through a heteroscedastic output layer. (Shukla & Marlin, 2022). MGP-TCN combines multi-task Gaussian Process to manage non-uniform sampling frequencies with temporal convolution network to capture temporal dependencies (Moor et al., 2019). However, these methods lack a representation learning paradigm and often struggle to capture evolving dynamics in partially-observed data. In contrast, our TrajGPT can be interpreted as discretized ODEs, allowing it to learn continuous dynamics via large-scale pre-training. Moreover, TrajGPT utilizes interpolation and extrapolation techniques from the neural ODE family to predict accurate trajectories.

## 3 METHODOLOGY

We denote an irregularly-sampled time series as $x = \{(x_1, t_1), \ldots, (x_N, t_N)\}$, where $N$ is the total number of samples. Each sample $(x_n, t_n)$ consists of an observation $x_n$ (e.g., a structured diagnosis code) and its associated timestamp $t_n$. The notations of variables are defined in Appendix. A.

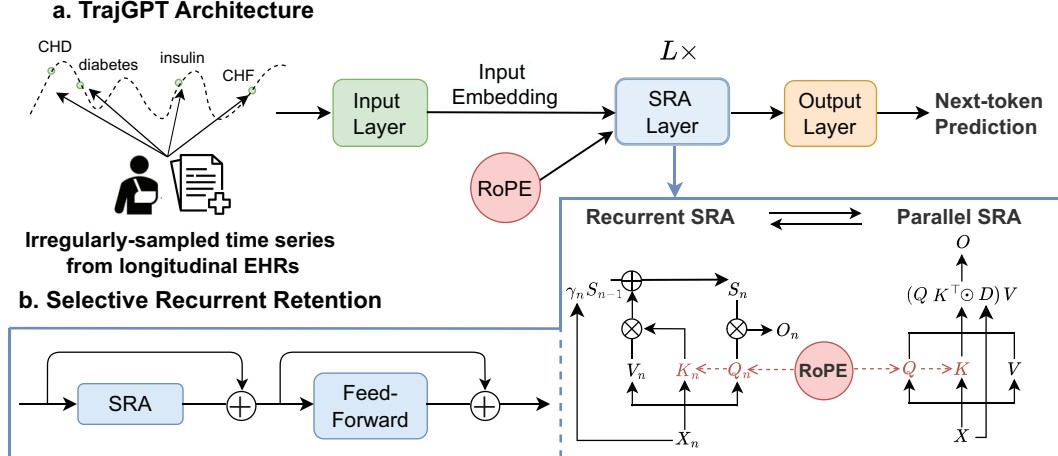

Figure 1: TrajGPT overview. **(a)**. TrajGPT processes irregularly-sampled time series by embedding an input sequence with RoPE. **(b)**. Each SRA layer comprises an SRA module and a feed-forward layer, with the SRA module capable of operating in both recurrent and parallel forms.

## 3.1 TRAJGPT METHODOLOGY

In the TrajGPT architecture illustrated in Fig. 1.a, each input sequence $x$ is first projected onto a token embedding $\boldsymbol{X} \in \mathbb{R}^{N \times d}$, where $N$ and $d$ denote the number of tokens and embedding size, respectively. A Rotary Position Embedding (RoPE) is then added to the token embedding, encoding relative positional information between tokens $n$ and $m$ (Su et al., 2022). Specifically, RoPE handles varying time intervals by encoding its relative distance $t_n - t_m$:

$$\boldsymbol{Q}_n = \boldsymbol{X}_n \boldsymbol{W}_Q e^{i\theta t_n}, \ \boldsymbol{K}_m = \boldsymbol{X}_m \boldsymbol{W}_K e^{-i\theta t_m}, \ \boldsymbol{V}_m = \boldsymbol{X}_m \boldsymbol{W}_V. \tag{1}$$

The resulting input embedding is then passed through $L$ SRA layers, each comprising an SRA module and a feed-forward layer. SRA module operates in either parallel or recurrent forms. In the *recurrent* forward pass, SRA computes the output representation $\boldsymbol{O}_n$ based on a state variable $\boldsymbol{S}$:

$$\boldsymbol{S}_n = \gamma_n \boldsymbol{S}_{n-1} + \boldsymbol{K}_n^\top \boldsymbol{V}_n, \ \boldsymbol{O}_n = \boldsymbol{Q}_n \boldsymbol{S}_n, \ \text{where } \gamma_n = \text{Sigmoid}(\boldsymbol{X}_n \mathbf{w}_\gamma^\top)^{\frac{1}{\tau}}. \tag{2}$$

The *data-dependent* decay $\gamma_n \in (0, 1]$ and learnable decay vector $\mathbf{w}_\gamma \in R^{1 \times d}$ enable SRA to selectively forget irrelevant past information based on contexts. For chronic diseases, TrajGPT assigns higher $\gamma_n$ values to slow down forgetting and capture long-term dependencies. Conversely, lower $\gamma_n$ values accelerate decay and prioritize recent events, making it more responsive to acute conditions. To avoid rapid decay from small $\gamma_n$ values, we introduce a temperature parameter $\tau = 20$ to help preserve information over long sequences. Given an initial state $\boldsymbol{S}_0 = 0$, we can rewrite the recurrent form in Eq. 2 in a *parallel* form as:

$$\boldsymbol{O} = (\boldsymbol{Q}\boldsymbol{K}^\top \odot \boldsymbol{D})\boldsymbol{V}, \ \boldsymbol{D}_{nm} = \begin{cases} \frac{b_n}{b_m}, & n \geq m \\ 0, & n < m \end{cases} \tag{3}$$

where $b_n = \prod_{t=1}^n \gamma_t$ indicates the cumulative decay term for token $n$, and $b_n/b_m$ captures the relative decay between tokens $n$ and $m$. We detail the equivalence between recurrence and parallelism in Appendix B. To capture a broader range of contexts, We extend the single-head SRA in Eq. 2 to a multi-head SRA:

$$\boldsymbol{O}_n^h = \boldsymbol{Q}_n^h \boldsymbol{S}_n^h, \ \boldsymbol{S}_n^h = \gamma_n^h \boldsymbol{S}_{n-1}^h + \boldsymbol{K}_n^{h\top} \boldsymbol{V}_n^h, \ \text{where } \gamma_n^h = \text{Sigmoid}(\boldsymbol{X}_n \mathbf{w}_\gamma^{h\top})^{\frac{1}{\tau}}, \tag{4}$$

Head-specific decay $\gamma_n^h$ adjusts the influence of past information based on contexts, with $\boldsymbol{w}_\gamma^h$ encoding different aspects of medical expertise.

## 3.2 TRAJGPT AS DISCRETIZED ODES

In this section, we establish theoretical connection between our proposed SRA module and ODEs. The recurrent form of SRA in Eq. 2 is a discretization of continuous-time ODE using zero-order

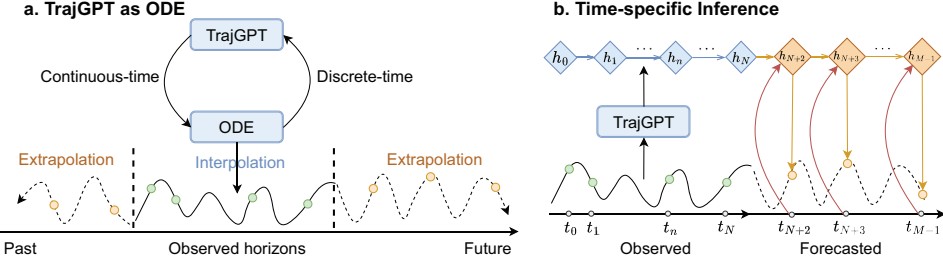

Figure 2: TrajGPT as discretized ODEs. **(a)**. TrajGPT performs interpolation and extrapolation by modeling continuous dynamics as discretized ODEs. **(b)**. Time-specific inference directly predicts irregular samples using previous hidden states and target timesteps.

hold (ZOH) rule (Gu et al., 2022). Appendix C provides a detailed proof establishing the theoretical connection that our model represents a discretized ODE. Appendix D provides a mathematical derivation of the ZOH discreization of continuous-time ODE, leading to our recurrent SRA module.

Given a first-order ODE, we can derive our recurrent SRA (Eq. 2) using a ZOH discretization with a discrete step size $\Delta$:

$$\boldsymbol{S}'(t) = \boldsymbol{A}\boldsymbol{S}(t) + \boldsymbol{B}\boldsymbol{X}(t), \ \boldsymbol{O}(t) = \boldsymbol{C}\boldsymbol{S}(t)$$

$$\text{where } \boldsymbol{A} = \frac{\ln(\boldsymbol{\Lambda}_t)}{\Delta}, \ \boldsymbol{B} = \boldsymbol{A}(e^{\Delta\boldsymbol{A}} - \boldsymbol{I})^{-1}\boldsymbol{K}_t^\top, \ \boldsymbol{C} = \boldsymbol{Q}_t, \ \boldsymbol{\Lambda}_t = \text{diag}(\boldsymbol{1}\gamma_t). \tag{5}$$

This ODE naturally models the continuous dynamics underlying irregularly-sampled data, with $\Delta$ corresponding to the varying time intervals between observations. Since the parameters $(\boldsymbol{A}, \boldsymbol{B}, \boldsymbol{C})$ depend on the $t$-th observation $\boldsymbol{X}(t)$, this continuous-time model becomes a neural ODE, $\boldsymbol{S}'(t) = f(\boldsymbol{S}(t), t, \theta_t)$, with a differentiable neural network $f$ and data-dependent parameters $\theta_t = (\boldsymbol{A}, \boldsymbol{B}, \boldsymbol{C})$ (Chen et al., 2018). Consequently, a single-head SRA serves as a discretized ODE with data-dependent parameters (i.e., neural ODE). TrajGPT with multi-head SRA operates as discretized ODEs, where each head corresponds to its own ODE and captures distinct dynamics.

As illustrated in Fig. 2.a, TrajGPT functions as discretized ODEs, enabling both interpolation and extrapolation of irregular time-series data. By capturing the underlying continuous dynamics, TrajGPT handles irregular input through discretization with varying step sizes. For interpolation, it simply evolves the dynamics within the observed timeframe using a unit discretization step size. For extrapolation, it evolves the dynamics forward or backward in time beyond the observed timeframe. Additionally, TrajGPT estimates disease risk trajectories by computing token probabilities at specific timesteps and evolving the dynamics through interpolation and extrapolation. A detailed trajectory analysis is provided in Section 5.3.

At inference time, we explore two strategies for forecasting irregularly-sampled time series: auto-regressive and time-specific inference (Fig. 2.b). Auto-regressive inference, commonly used by standard Transformer models, makes sequential predictions at equal intervals and selects the target timesteps accordingly. Given that TrajGPT functions as discretized ODEs, we introduce a novel *time-specific inference* to predict at arbitrary timesteps. To forecast a target time point $(x_{n'}, t_{n'})$, TrajGPT utilizes both the target timestep $t_{n'}$ and the last observation $(x_n, t_n)$ to predict the corresponding observation $x_{n'}$. It calculates the target output representation $\boldsymbol{O}_{n'} = \boldsymbol{Q}_{n'}\boldsymbol{S}_{n'}$, taking into account the discrete step size $\Delta t_{n',n} = t_{n'} - t_n$ and the updated state $\boldsymbol{S}_{n'} = \boldsymbol{D}_{\Delta t_{n',n}}\boldsymbol{S}_n + \boldsymbol{K}_n^\top\boldsymbol{V}_n$.

### 3.3 COMPUTATIONAL COMPLEXITY

TrajGPT with its efficient SRA mechanism achieves linear training complexity of $O(N)$ and constant inference complexity of $O(1)$ with respect to sequence length $N$. In contrast, standard Transformer models incur quadratic training complexity of $O(N^2)$ and linear inference complexity of $O(N)$ (Katharopoulos et al., 2020). This computational bottleneck arises from the vanilla self-attention mechanism, where $\text{Attention}(\boldsymbol{X}) = \text{Softmax}(\boldsymbol{Q}\boldsymbol{K}^T)\boldsymbol{V}$, resulting in a training complexity of $O(N^2d)$. When dealing with long sequences (i.e., $N >> d$), the quadratic term $O(N^2)$ becomes a bottleneck for standard Transformer models.

As a variant of linear attention (Katharopoulos et al., 2020), the SRA mechanism in TrajGPT overcomes this quadratic bottleneck of standard Transformer, achieving linear training complexity for long sequences. In recurrent SRA, $\boldsymbol{O}_n = \boldsymbol{Q}_n \boldsymbol{S}_n, \boldsymbol{S}_n = \gamma_n \boldsymbol{S}_{n-1} + \boldsymbol{K}_n^\top \boldsymbol{V}_n$, both $\boldsymbol{Q}_n \boldsymbol{S}_n$ and $\boldsymbol{K}_n^\top \boldsymbol{V}_n$ have $O(d^2)$ complexity. By recursively updating over $N$ tokens, the total complexity becomes $O(Nd^2)$. For inference, TrajGPT proposes auto-regressive and time-specific methods. The auto-regressive inference sequentially generates sequences with equally spaced time intervals like the GPT model, incurring linear complexity of $O(N)$. In contrast, time-specific inference directly predicts the target time point with constant complexity of $O(1)$. Thus, TrajGPT achieves $O(N)$ training complexity and $O(1)$ inference complexity, making it computationally efficient for long sequences.

## 4 EXPERIMENTAL DESIGN

### 4.1 DATASET AND PRE-PROCESSING

Population Health Record (PopHR) database hosts massive amounts of longitudinal claim data from the provincial government health insurer in Quebec, Canada on health service use (Shaban-Nejad et al., 2016; Yuan et al., 2018). In total, there are approximately 1.3 million participants in the PopHR database, representing a randomly sampled 25% of the population in the metropolitan area of Montreal between 1998 and 2014. Cohort memberships are maintained dynamically by removing deceased residents and actively enrolling newborns and immigrants. We extracted irregularly-sampled time series from the PopHR database. Specifically, we converted ICD-9 diagnostic codes to integer-level phenotype codes (PheCodes) using the PheWAS catalog (Denny et al., 2013; 2010). We selected 194 unique PheCodes, each with over 50,000 occurrences. We excluded patients with fewer than 50 PheCode records, resulting in a final dataset of 489,000 patients, with an average of 112 records per individual. The dataset was then split into training (80%), validation (10%), and testing (10%) sets.

The eICU Collaborative Research Database is a multi-center intensive care unit (ICU) database containing over 200,000 admissions from ICUs monitored by eICU programs in the United States. It offers de-identified EHR data, encompassing patient demographics, diagnoses, treatments, and interventions. To extract irregularly-sampled time series, we convert ICD-9 codes to 288 integer-level PheCodes. We harmonized drugs with the same identity but differing names and dosages, resulting in 228 unique drugs. We performed representation learning with a 15-minute interval for clinical events (diagnosis and drug). This resulted in a final dataset of 139,367 patients, with an average of 19 drugs and 3 ICD codes per patient.

### 4.2 POPHR EXPERIMENT DESIGN

**Forecast irregular diagnostic codes** We evaluated the long-term forecasting task using a look-up window of 50 time points (e.g., diagnosis codes) to predict the remaining codes in the forecasting windows. We measured model performance using the top-$K$ recall with $K = (5, 10, 15)$. This metric mimics the behavior of doctors conducting differential diagnosis, where they list the most probable diagnoses based on a patient's symptoms Choi et al. (2016). Since next-token prediction is inherently forecasting, TrajGPT enables zero-shot forecasting without requiring fine-tuning.

**Drug usage prediction** In this application, we predict whether each diabetic patient started insulin treatment within 6 months of their initial diabetes diagnosis. Following the preprocessing from previous work (Song et al., 2021), we extracted 78,712 diabetic patients with PheCode 250, where 11,433 patients were labeled as positive. Due to class imbalance, we use the area under precision-recall curve (AUPRC) as the evaluation metric. To avoid information leakage, we truncated sequence representations at the first diabetes record. To assess generalizability, we performed zero-shot classification, few-shot classification with 5 samples, and fine-tuning on the full dataset.

**Phenotype classification** PopHR database provides rule-based labels for congestive heart failure (CHF), with 3.2% of the total population labeled as positive. Given the class imbalance, we utilize the AUPRC metric to evaluate performance on the rare positive class. To assess the generalizability of the pre-trained TrajGPT, we conducted zero-shot classification, few-shot classification with 5 samples, and fine-tuning on the entire dataset.

### 4.3 EICU EXPERIMENT DESIGN

**Forecast irregular diagnoses and drugs** We conducted the forecasting task using a look-up window of 10 time points to predict the remaining codes in the forecasting windows. We assessed forecasting performance using the top-$K$ recall with $K = (10, 20)$.

**Early Detection of Sepsis** We defined a 72-hour observation period following ICU admission. We identified patients without sepsis during the first 8 hours and predict sepsis onset in the remaining windows. This task was performed using both zero-shot learning and fine-tuning on the full dataset.

### 4.4 BASELINES

For PopHR dataset, we compared our model against several time-series transformer baselines, including TimelyGPT (Song et al., 2024a), BiTimelyGPT (Song et al., 2024b), Informer (Zhou et al., 2021), Fedformer (Zhou et al., 2022), AutoFormer (Wu et al., 2021), PatchTST (Nie et al., 2023), TimesNet (Wu et al., 2023), ContiFormer (Chen et al., 2024), PrimeNet (Chowdhury et al., 2023), and Mamba (Gu & Dao, 2024; Dao & Gu, 2024). BiTimelyGPT and PatchTST are encoder-only models that require fine-tuning for forecasting tasks, while other Transformer models with decoders can forecast without additional fine-tuning. We also evaluated models designed for irregularly-sampled time series, including mTAND (Shukla & Marlin, 2021), GRU-D (Che et al., 2018), RAINDROP (Zhang et al., 2022), SeFT (Horn et al., 2020), ODE-RNN (Rubanova et al., 2019), HeTVAE (Shukla & Marlin, 2022), and MGP-TCN (Moor et al., 2019). For eICU dataset, we compared TrajGPT against efficient models from Section 5.2, including TimelyGPT, PatchTST, TimesNet, ContiFormer, PrimeNet, Mamba-2, MTand, and SeFT. Since these models do not have a pre-training method, they were trained from scratch on the training set. We followed previous works to set Transformer parameters to about 7.5 million (Table 5).

**Transformer Pre-training paradigm** With a cross-entropy loss, TrajGPT employs a next-token prediction task to pre-train generalizable temporal representations from unlabeled data (Radford et al., 2019). Given a sequence with a [SOS] token, TrajGPT predicts subsequent tokens by shifting the sequence to the right. The output representation of each token is fed into a linear layer for next-token prediction. For other models without an established pre-training paradigm, we employed a masking-based method by randomly masking 40% of timesteps with zeros (Zerveas et al., 2021). All Transformer models performed 20 epochs of pre-training with cross-entropy loss. When fine-tuning was applicable, we performed 5 epochs of end-to-end fine-tuning on the entire model.

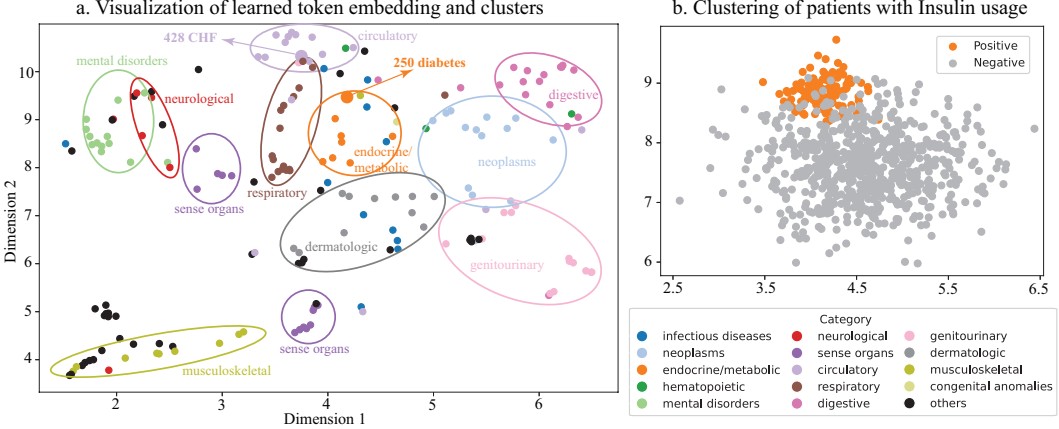

Figure 3: Visualization of token embeddings and sequence representations. **(a)**. Visualization of token embeddings across 15 disease categories, where token nodes are colored and clustered by categories. **(b)**. Visualization of sequence representations for diabetic patients, highlighting insulin usage within six months of diagnosis. The distinction of two classes enables zero-shot classification.

## 5 RESULTS

### 5.1 QUALITATIVE ANALYSIS OF EMBEDDINGS

In this section, we provided a qualitative analysis of the token embeddings and sequence representations learned by our TrajGPT on the PopHR database (Fig. 3). We applied Uniform Manifold Approximation and Projection (UMAP) to visualize the global token embedding, with nodes colored and clustered by disease categories. The results reveal 12 clearly separated clusters. Some nodes are projected into other categories but still reflect meaningful clinical relationships; for instance, the mental disorders cluster (in green color) includes a black dot representing adverse drug events and drug allergies, implying high risk of opioid usage among the psychiatric group (Zhu et al., 2021). Related disease categories with clinical relevance tend to cluster near each other. For example, mental disorders are closely clustered with neurological diseases, and circulatory diseases are adjacent to endocrine/metabolic diseases. We visualized the projected head-specific decay vectors $\boldsymbol{w}_\gamma^h$ in Eq. 4 using the UMAP techniques (Fig. ??). It shows that the eight decay vectors are projected into distinct 2-D vectors, indicating that they capture different patterns.

In Fig. 3.b, we visualize the sequence representations to demonstrate TrajGPT's ability to perform zero-shot classification of initial insulin usage among diabetic patients. To prevent information leakage, the sequence representations were truncated at the first diabetes record. These sequence representations were projected onto the same scale as the token embeddings in Fig. 3.a, allowing for direct comparison with the disease clusters. Patients taking future insulin treatment have embeddings closely aligned with the endocrine/metabolic cluster, indicating a strong association with diabetes-related conditions. In contrast, non-insulin patients are dispersed across various clusters, suggesting less severe diabetes histories. The clear separation between these groups highlights TrajGPT's ability to perform zero-shot classification, showcasing the generalizability of its learned representations.

### 5.2 QUANTITATIVE RESULTS ON PopHR DATASET

TrajGPT with time-specific inference achieves the highest recall at $K = 10$ and $K = 15$, with scores of 71.7% and 84.1%, respectively (Table 1). At $K = 5$, TrajGPT achieves the second-highest recall with 57.4%. Notably, time-specific inference outperforms the auto-regressive inference approach, demonstrating its effectiveness in forecasting based on the learned continuous dynamics. These results highlight TrajGPT's strength in pre-training on underlying dynamics from sparse and irregular time-series data, facilitating accurate trajectory forecasting over irregular time intervals.

We then examined the distributions of top-10 recall across three forecast windows, comparing the two inference methods of TrajGPT as well as TimelyGPT, PatchTST, and mTAND (Fig. 6). TrajGPT's time-specific inference consistently outperforms auto-regressive inference as the forecasting window increases, as it accounts for evolving states and query timesteps over irregular intervals. As expected, all models experience a performance decline as the forecast window increases, reflecting the increased uncertainty in long-term trajectory prediction. Despite this, TrajGPT achieves superior and more stable performance within the first 100 steps. In comparison, PatchTST shows a drastic decline as the window size increases, reflecting its difficulty with extrapolation. Therefore, TrajGPT excels in forecasting health trajectories through its time-specific inference.

We evaluated two classification tasks—insulin usage prediction and CHF phenotype classification—under three settings: zero-shot learning, few-shot learning with $S = 5$ samples, and fine-tuning on the entire dataset. Notably, non-Transformer models designed for irregularly-sampled time series (i.e., the last five methods in Table. 1) were trained from scratch. The results are summarized in Table. 1. For classification tasks, TrajGPT achieves the highest zero-shot results, with 67.2% for insulin and 72.8% for CHF. This success can be attributed to TrajGPT's ability to learn distinct clusters of sequence representations, as discussed in Section 5.1. For 5-shot classification, TrajGPT achieves the second-best performance in both tasks. For fine-tuning, it obtains the second best performance of 83.9% in insulin prediction, only 0.3% behind the best-performing BiTimelyGPT. We also compared TrajGPT with algorithms specifically designed for irregularly-sampled time series. These methods generally perform worse in insulin usage prediction, likely due to their difficulty in capturing meaningful temporal dependencies from truncated sequences. However, mTAND outperforms all models in the CHF task, achieving the best result at 85.4%.

Table 1: The quantitative results on the diagnosis forecasting, insulin usage, and CHF classification. performance on PopHR dataset. Metrics are reported as average (standard error) from a bootstrap evaluation of variance. The bold and underline indicate the best and second best results, respectively. $S$ indicates the number of few-shot examples. $-$ indicates non-applicable.

| Methods / Tasks (%) | Forecasting | | | Diabetes-Insulin | | | CHF | | |
|---|---|---|---|---|---|---|---|---|---|
| | K = 5 | 10 | 15 | S = 0 | 5 | all | S = 0 | 5 | all |
| TrajGPT (Time-specific) | 57.4 (3.2) | **71.7 (2.6)** | **84.1 (2.4)** | **67.2 (3.1)** | 70.2 (3.0) | 75.5 (2.6) | **72.8 (2.4)** | 75.9 (2.1) | 83.9 (2.0) |
| TrajGPT (Auto-regressive) | 53.3 (3.9) | 65.5 (3.4) | 77.2 (2.7) | — | — | — | — | — | — |
| TimelyGPT | **58.2 (3.7)** | 70.3 (3.1) | 82.1 (2.5) | 58.2 (2.8) | 64.4 (2.5) | 70.7 (2.6) | 66.9 (2.3) | 71.0 (2.2) | 81.2 (2.0) |
| BiTimelyGPT | 48.2 (3.3) | 63.3 (3.2) | 70.5 (2.8) | 65.3 (3.1) | **70.8 (2.9)** | **75.8 (3.0)** | 70.4 (2.4) | 74.5 (2.3) | 83.8 (2.1) |
| Informer | 46.4 (2.9) | 60.1 (2.8) | 71.2 (2.6) | 62.1 (4.6) | 66.2 (4.5) | 71.5 (3.8) | 62.9 (4.2) | 67.4 (3.9) | 80.8 (3.5) |
| Autoformer | 42.9 (2.9) | 57.4 (2.7) | 68.6 (2.4) | 63.5 (3.8) | 66.8 (3.6) | 72.7 (3.4) | 65.3 (3.5) | 69.6 (3.7) | 81.6 (3.2) |
| Fedformer | 43.3 (2.7) | 58.3 (2.5) | 69.6 (2.4) | 64.2 (4.3) | 68.4 (4.2) | 73.1 (3.8) | 68.2 (3.8) | 69.8 (3.5) | 81.9 (2.9) |
| PatchTST | 48.2 (2.7) | 65.5 (2.4) | 73.3 (2.2) | 66.8 (2.6) | 69.7 (2.7) | 75.1 (2.4) | 72.2 (2.3) | **76.3 (1.9)** | **84.2 (2.1)** |
| TimesNet | 46.5 (3.7) | 64.3 (3.0) | 71.5 (2.5) | 64.2 (3.2) | 67.9 (2.8) | 72.8 (2.9) | 67.8 (3.1) | 72.5 (3.0) | 82.6 (2.8) |
| ContiFormer | 52.8 (3.1) | 67.2 (2.8) | 76.9 (2.5) | 63.5 (3.3) | 68.0 (3.1) | 75.0 (2.9) | 68.4 (2.4) | 74.9 (2.2) | 83.1 (2.3) |
| PrimeNet | 52.5 (3.2) | 69.7 (2.8) | 81.8 (2.3) | 65.6 (3.0) | 69.5 (2.9) | 73.8 (2.7) | 71.5 (2.7) | 75.5 (2.9) | 84.0 (2.4) |
| Mamba-1 | 46.5 (3.6) | 62.4 (3.1) | 73.6 (2.6) | 61.5 (3.6) | 67.4 (3.2) | 72.5 (3.0) | 65.2 (3.1) | 70.1 (2.9) | 81.4 (2.4) |
| Mamba-2 | 51.4 (3.2) | 69.8 (2.9) | 80.7 (2.5) | 64.6 (3.1) | 69.9 (2.8) | 74.8 (2.4) | 69.6 (2.7) | 73.9 (2.8) | 83.4 (2.3) |
| MTand | 52.6 (2.8) | 70.2 (2.5) | 83.7 (1.9) | — | — | 74.6 (3.1) | — | — | **85.4 (2.5)** |
| GRU-D | 54.2 (4.0) | 69.5 (3.4) | 80.5 (3.1) | — | — | 72.1 (3.2) | — | — | 79.9 (2.7) |
| RAINDROP | 46.5 (2.9) | 67.2 (2.5) | 72.2 (2.2) | — | — | 70.5 (2.8) | — | — | 82.4 (2.4) |
| SeFT | 49.3 (2.6) | 68.1 (2.2) | 79.4 (1.7) | — | — | 71.7 (2.6) | — | — | 83.4 (2.3) |
| ODE-RNN | 54.7 (4.2) | 70.6 (3.5) | 78.6 (2.8) | — | — | 73.5 (3.6) | — | — | 82.9 (3.0) |
| HeTVAE | 51.1 (3.9) | 70.1 (3.4) | 83.2 (3.2) | — | — | 71.4 (3.6) | — | — | 81.6 (3.2) |
| MGP-TCN | 43.5 (3.5) | 57.2 (3.1) | 69.1 (2.9) | — | — | 73.9 (3.6) | — | — | 82.4 (3.5) |

## 5.3 TRAJECTORY ANALYSIS

In this analysis, we aimed to demonstrate TrajGPT's effectiveness in trajectory modeling and provide insights into its classification performance. To achieve this, we conducted case studies on two patients: one diagnosed with diabetes and another with CHF. We visualized the observed and predicted disease trajectories for both patients, along with estimated risk trajectories over their lifetimes. As discussed in Section 3.2, we interpolated risks within the observed timeframe and extrapolated beyond it in both directions, computing risk as the token probability at each timestep. We also calculated risk growth by comparing each timestep to the previous one, identifying the ages with high risk growth as well as the associated phenotypes. By comparing disease and risk trajectories, we evaluated phenotype progression, disease comorbidity, and long-term risk development.

In Fig. 4.a, TrajGPT with time-specific inference achieves a top-10 recall of 90.1% for this diabetic patient. TrajGPT accurately predicts most diseases in the endocrine/metabolic and circulatory systems. Although this patient has no prior diabetes diagnosis in the observed data, TrajGPT successfully forecasts diabetes onset by identifying related metabolic and circulatory symptoms. Fig. 4.b illustrates the predicted risk trajectory for this patient, indicating a gradual increase in diabetes risk with age. We highlight specific phenotypes that contribute to the noticeable high risk growth between ages 59 and 62, including chronic IHD, hypothyroidism, obesity, and arrhythmia (Biondi et al., 2019). These conditions are common comorbidities of diabetes, substantially elevating the likelihood of diabetes onset over time. In Fig. 4.c, we visualize the disease trajectory of a CHF patient, for whom TrajGPT produces a top-10 recall of 84.7%. TrajGPT accurately predicts a broad range of circulatory, respiratory, and endocrine/metabolic diseases. Despite the absence of prior CHF diagnosis, TrajGPT successfully predicts the onset of CHF based on a series of related circulatory conditions Correale et al. (2020). In Fig. 4.d, the predicted risk trajectory reveals two spikes in risk growth at ages 65 and 74, corresponding to successive occurrences of circulatory diseases (Khan et al., 2020). This analysis demonstrates TrajGPT's ability to forecast unseen phenotypes based on disease comorbidity and the risking risk with age. As a result, TrajGPT's ability to model health trajectories and capture disease progression enhances its classification performance.

The ability to forecast diagnostic codes highlights the potential of Transformer models for health trajectory analysis. These codes can serve a broad range of administrative purposes, such as estimating the diagnostic related group (DRG) for inpatients to improve the efficiency and quality of inpatient care (Renc et al., 2024). They also hold significant potential for informing clinical care, including directing the need for preventive care and identifying potential complications (Shankar et al., 2023).

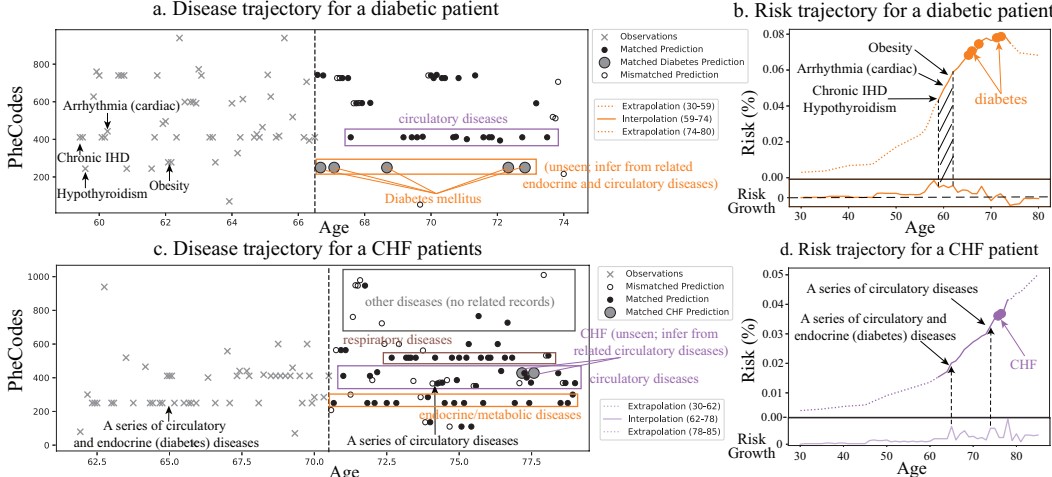

Figure 4: Predicted health trajectories for a diabetic patient (top) and a CHF patient (bottom). **Panels (a) and (b)** show the inferred disease trajectories with look-up and forecast windows. Matched predictions (solid circles) occur when the top 10 predicted PheCodes match the ground-truth. Larger solid circles indicate correctly predicted diabetes or CHF. **Panels (c) and (d)** display the predicted risk trajectories, showing increasing risks with age. For a target disease, TrajGPT computes risk as the token probability at each timestep and calculates risk growth as the difference between consecutive values. We highlight key timesteps to indicate significant risk growth and the associated phenotypes.

Table 2: We evaluate TrajGPT and baselines on the eICU dataset for the event forecasting and sepsis prediction. TrajGPT achieves top performance in both clinical event forecasting tasks and zero-shot classification of sepsis. Metrics are reported as average (standard error) from a bootstrap evaluation of variance. The bold and underline indicate the best and second best results, respectively. $S$ indicates the number of few-shot examples. $-$ indicates non-applicable.

| Methods / Tasks (%) | Forecasting | | Sepsis | |
|---|---|---|---|---|
| | K = 10 | 20 | S=0 | All |
| TrajGPT (Time-specific) | **57.8 (2.9)** | **69.3 (2.1)** | **45.1 (2.7)** | 51.3 (2.4) |
| TrajGPT (Auto-regressive) | 54.1 (3.2) | 64.9 (2.3) | — | — |
| TimelyGPT | 56.9 (3.2) | 67.1 (2.4) | 42.0 (2.5) | 48.5 (2.2) |
| PatchTST | 55.2 (2.7) | 66.0 (1.7) | 44.5 (2.2) | 51.8 (1.8) |
| TimesNet | 52.9 (3.1) | 60.3 (2.3) | 41.2 (3.1) | 47.5 (2.6) |
| ContiFormer | 57.1 (2.2) | 66.8 (2.2) | 41.7 (2.5) | 50.6 (2.8) |
| PrimeNet | 53.4 (2.3) | 67.5 (2.0) | 44.0 (2.3) | 51.2 (1.9) |
| Mamba-2 | 55.7 (2.8) | 65.2 (2.3) | 43.6 (2.8) | 49.5 (2.3) |
| MTand | 53.9 (2.4) | 67.4 (1.6) | — | **52.5 (2.1)** |
| ODE-RNN | 55.7 (3.4) | 67.8 (2.8) | — | 49.2 (2.9) |

## 5.4 QUANTITATIVE RESULTS ON eICU DATASET

For the eICU datasets, we evaluated TrajGPT on irregular clinical event forecasting (diagnoses and drugs) and early detection of sepsis, with results summarized in Table. 2. Note that the recall values for the joint prediction of diagnoses and drugs are lower due to the larger hypothesis space for this task Choi et al. (2016). Despite the increased complexity compared to predicting diagnoses alone, TrajGPT with time-specific inference achieved superior performance over baseline models, resulting in a top-10 recall of 57.8% and a top-20 recall of 69.3%. This superior performance can be attributed to the effectiveness of time-specific inference, which improve top-10 and top-20 recall rates by 3.7% and 4.4% respectively, compared to auto-regressive inference. The representation learning methods designed specifically for irregularly-sampled time series demonstrated better overall performance. Additionally, ODE-RNN achieves the second-best performance with a top-20 recall of 67.8%. These findings highlight that both TrajGPT's time-specific inference and ODE-RNN leverage the strengths

Table 3: Ablation results of TrajGPT by selectively removing components and comparing inference methods. Performance is evaluated on forecasting task with a the of top-10 recall.

| Forecast irregular diagnosis codes ($K$=10) | Time-specific inference | Auto-regressive inference |
|---|---|---|
| TrajGPT | **71.7** | 65.5 |
|    w/o decay gating (i.e., fixed $\gamma$) | 70.3 | 64.0 |
|       w/o RoPE (i.e., absolute PE) | 67.8 | 63.2 |
|          w/o linear attention (i.e., GPT-2) | — | 61.2 |
| TrajGPT (without Pre-training) | 67.1 | ? |

of modeling underlying dynamics to enhance forecasting accuracy. For the sepsis prediction task, TrajGPT outperforms all baselines in the zero-shot setting, achieving an AUPRC of 45.1%. While MTand performs best when trained from scratch, its reliance on a bespoke shallow model targeting a single outcome limits its scalability and applicability in clinical settings. In summary, TrajGPT leverages pre-trained generalizable patterns to enable zero-shot learning, effectively detecting early sepsis without additional training.

## 5.5 ABLATION STUDY

To evaluate the contributions of key components in TrajGPT, we performed ablation studies by selectively removing components such as decay gating, RoPE, and the linear attention module. We compared the time-specific inference and auto-regressive inference under different ablation setups. Notably, removing all components results in a vanilla GPT-2, which can only perform auto-regressive inference. The ablation studies were assessed on the forecasting task using the top-10 recall metric.

As shown in Table 3, removing the data-dependent decay and RoPE results in performance declines of 1.4% and 2.5%, respectively. This highlights the critical role of these modules in handling irregular time intervals by prioritizing recent data while attenuating the influence of distant ones. Replacing time-specific inference with auto-regressive inference leads to performance drops ranging from 4.6% to 6.2%, with the most significantly drop in TrajGPT. Furthermore, vanilla GPT-2 with auto-regressive inference produces the lowest performance, falling behind TrajGPT with time-specific inference by 10.5%. Time-specific inference uses varied time intervals for a single inference, reducing both computational steps and error accumulation for better performance.

## 6 CONCLUSION AND FUTURE WORK

The current paradigm in clinical practice relies on bespoke shallow models targeting single outcomes, highlighting the need for models capable of predicting diverse patient outcomes with minimal or no refinement (Moor et al., 2023). Developing such models for healthcare has to account for the irregular sampling of medical records, as improper modeling can lead to faulty inferences (Agniel et al., 2018). Our research proposes a novel architecture, TrajGPT, designed for irregular time-series representation learning and health trajectory analysis. To achieve this, TrajGPT introduces an SRA mechanism with a data-dependent decay, allowing the model to selectively forget irrelevant past information based on contexts. By interpreting TrajGPT as discretized ODEs, it effectively captures the continuous dynamics underlying irregularly-sampled time series, enabling both interpolation and extrapolation. For the forecasting task, TrajGPT provides an effective time-specific inference by evolving the dynamics according to varying time intervals. TrajGPT demonstrates strong zero-shot performance across multiple tasks, including diagnosis forecasting, drug usage prediction, and phenotype classification. TrajGPT also provides interpretable trajectory analysis, aiding clinicians in understanding the extrapolated disease progression along with risk growth.

To further validate generalizability, we will compare TrajGPT with foundation LLMs, such as GPT-based (Wang et al., 2024) and Llama-based (Rasul et al., 2024) models. Our work currently focuses on irregularly-sampled time series with discrete data (i.e., diagnoses and drugs); we plan to expand this to continuous multivariate time series, such as ICU measurements Johnson et al. (2023). While we focus on in-domain data, we will explore representation learning and trajectory analysis on out-of-distribution data as future works.

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

# A DENOTATIONS OF VARIABLES

Table 4: Notations in TrajGPT

| Notations | Descriptions | Notations | Descriptions |
|---|---|---|---|
| $x = \{(x_1, t_1), \ldots, (x_N, t_N)\}$ | An irregularly-sample time series | $N$ | Number of tokens |
| $x_n$ | An observation | $t_n$ | Corresponding timestamp |
| $\boldsymbol{X} \in \mathbb{R}^{N \times d}$ | A sequence of tokens | $d$ | Hidden dimension |
| $L$ | Number of layers | $H$ | Number of Heads |
| $\boldsymbol{Q}, \boldsymbol{K}, \boldsymbol{V} \in \mathbb{R}^{N \times d}$ | Query, key, value matrices | $\boldsymbol{W}_Q, \boldsymbol{W}_K, \boldsymbol{W}_V \in \mathbb{R}^{d \times d}$ | Projection matrices for $\boldsymbol{Q}, \boldsymbol{K}, \boldsymbol{V}$ |
| $\boldsymbol{O} \in \mathbb{R}^{N \times d}$ | Output embedding | $\boldsymbol{S} \in \mathbb{R}^{d \times d}$ | State variable |
| $\theta$ | Rotary angle hyperparameter | $\gamma \in (0, 1]$ | Data-dependent decay |
| $\mathbf{w}_\gamma \in \mathbb{R}^{d \times 1}$ | Decay weight vector | $\tau = 20$ | Temperature term |
| $b_n = \prod_{t=1}^{n} \gamma_t$ | Cumulative decay | $\boldsymbol{D} \in \mathbb{R}^{N \times N}$ | Decay matrix |

# B DERIVATION OF SRA LAYER

Starting from the recurrent form of the TrajGPT model in Eq. 2, we derive the state variable $S$ by assuming $\boldsymbol{S}_0 = 0$:

$$\boldsymbol{S}_n = \gamma_n \boldsymbol{S}_{n-1} + \boldsymbol{K}_n^\top \boldsymbol{V}_n$$

$$\boldsymbol{S}_1 = \boldsymbol{K}_1^\top \boldsymbol{V}_1$$

$$\boldsymbol{S}_2 = \gamma_2 \boldsymbol{K}_1^\top \boldsymbol{V}_1 + \boldsymbol{K}_2^\top \boldsymbol{V}_2$$

$$\boldsymbol{S}_3 = \gamma_3 \gamma_2 \boldsymbol{K}_1^\top \boldsymbol{V}_1 + \gamma_2 \boldsymbol{K}_2^\top \boldsymbol{V}_2 + \boldsymbol{K}_3^T \boldsymbol{V}_3$$

$$\vdots$$

$$\boldsymbol{S}_n = \sum_{m=1}^{n} \left( \prod_{t=m+1}^{n} \gamma_t \right) \boldsymbol{K}_m^\top \boldsymbol{V}_m = \sum_{m=1}^{n} \left( \frac{b_n}{b_m} \right) \boldsymbol{K}_m^\top \boldsymbol{V}_m, \text{ where } b_n = \prod_{t=1}^{n} \gamma_t, \tag{6}$$

where we get the generalized updates of $\boldsymbol{S}_n$ using the cumulative decay term $b_n = \prod_{t=1}^{n} \gamma_t$. We can compute the output representation $\boldsymbol{O}_n$ using $\boldsymbol{Q}_n$ and $\boldsymbol{S}_n$:

$$\boldsymbol{O}_n = \boldsymbol{Q}_n \boldsymbol{S}_n = \boldsymbol{Q}_n \sum_{m=1}^{n} \left( \frac{b_n}{b_m} \right) \boldsymbol{K}_m^\top \boldsymbol{V}_m. \tag{7}$$

To represent Eq. 7 in matrix form, we introduce a causal decay matrix $\boldsymbol{D}$, where each element $\boldsymbol{D}_{nm} = \prod_{t=m+1}^{n} \gamma_t$ represents the decay relationship between two tokens $n$ and $m$:

$$\boldsymbol{D} = \begin{bmatrix} \frac{b_1}{b_1} & 0 & \cdots & 0 \\ \frac{b_2}{b_1} & \frac{b_2}{b_2} & \cdots & 0 \\ \vdots & \vdots & \ddots & \vdots \\ \frac{b_N}{b_1} & \cdots & \cdots & \frac{b_N}{b_N} \end{bmatrix} = \begin{bmatrix} 1 & 0 & \cdots & 0 \\ \gamma_2 & 1 & \cdots & 0 \\ \vdots & \vdots & \ddots & \vdots \\ \prod_{t=2}^{N} \gamma_t & \cdots & \cdots & 1 \end{bmatrix}. \tag{8}$$

Using this decay matrix $\boldsymbol{D}$, we give the matrix form of the recurrent updates of $O_n$ in Eq. 7:

$$\boldsymbol{O}_n = \boldsymbol{Q}_n \sum_{m=1}^{n} \boldsymbol{D}_{nm} \boldsymbol{K}_m^\top \boldsymbol{V}_m$$

$$= \boldsymbol{Q}_n \left( \boldsymbol{D}_{n1} \boldsymbol{K}_1^\top \boldsymbol{V}_1 + \cdots + \boldsymbol{D}_{nn} \boldsymbol{K}_n^\top \boldsymbol{V}_n \right)$$

$$= \boldsymbol{Q}_n (\boldsymbol{D}_{n1} \boldsymbol{K}_1^\top \boldsymbol{V}_1 + \cdots + \boldsymbol{D}_{nn} \boldsymbol{K}_n^\top \boldsymbol{V}_n + \underbrace{\boldsymbol{D}_{n,n+1}}_{0} \boldsymbol{K}_{n+1}^\top \boldsymbol{V}_{n+1} + \cdots + \underbrace{\boldsymbol{D}_{nN}}_{0} \boldsymbol{K}_N^\top \boldsymbol{V}_N)$$

$$= \left( (\boldsymbol{Q}_n \boldsymbol{K}^\top) \odot \boldsymbol{D}_n \right) \boldsymbol{V}. \tag{9}$$

To express the computation of all tokens, we obtain the parallel form of SRA as follows:

$$\boldsymbol{O} = (\boldsymbol{Q} \boldsymbol{K}^\top \odot \boldsymbol{D}) \boldsymbol{V}, \quad \boldsymbol{D}_{nm} = \begin{cases} \frac{b_n}{b_m}, & n \geq m \\ 0. & n < m \end{cases}. \tag{10}$$

## C  TRAJGPT AS SSM AND NEURAL ODE

The continuous-time SSM defines a linear mapping from an $t$-step input signal $\boldsymbol{X}(t)$ to output $\boldsymbol{O}(t)$ via a state variable $\boldsymbol{S}(t)$. It is formulated as a first-order ODE:

$$\boldsymbol{S}'(t) = \boldsymbol{A}\boldsymbol{S}(t) + \boldsymbol{B}\boldsymbol{X}(t), \; \boldsymbol{O}(t) = \boldsymbol{C}\boldsymbol{S}(t), \tag{11}$$

where $\boldsymbol{A}, \boldsymbol{B}, \boldsymbol{C}$ denote the state matrix, input matrix, and output matrix respectively. Since data in real-world is typically discrete instead of continuous, continuous-time SSMs require discretization process to align with the sample rate of the data. With the discretization via zero-order hold (ZOH) rule (Gu et al., 2022), this continuous-time SSM in Eq. 11 becomes a discrete-time model:

$$\boldsymbol{S}_t = \bar{\boldsymbol{A}}\boldsymbol{S}_{t-1} + \bar{\boldsymbol{B}}\boldsymbol{X}_t, \; \boldsymbol{O}_t = \boldsymbol{C}\boldsymbol{S}_t$$
$$\bar{\boldsymbol{A}} = e^{\Delta \boldsymbol{A}}, \; \bar{\boldsymbol{B}} = (e^{\Delta \boldsymbol{A}} - \boldsymbol{I})\boldsymbol{A}^{-1}\boldsymbol{B}, \tag{12}$$

where $\bar{\boldsymbol{A}}$ and $\bar{\boldsymbol{B}}$ are the discretized matrices and $\Delta$ is the discrete step size. We provide a detailed derivation of ZOH discretization in Appendix D.

Here, we show that a single-head SRA module is a special case of the discrete-time SSM defined in Eq. 12, and then we derive its corresponding continuous-time SSM. To achieve it, we first rewrite the recurrent SRA (Eq. 2) as follows:

$$\boldsymbol{S}_t = \boldsymbol{\Lambda}_t \boldsymbol{S}_{t-1} + \boldsymbol{K}_t^{\top} \boldsymbol{V}_t,$$
$$\boldsymbol{O}_t = \boldsymbol{Q}_t \boldsymbol{S}_t, \tag{13}$$

where $\boldsymbol{\Lambda}_t = \mathrm{diag}(\mathbf{1}\gamma_t)$ is a diagonal matrix with all diagonal elements equal to $\gamma_t$. In this way, the recurrent form of SRA in Eq. 13 corresponds to the discrete-time SSM defined in Eq. 12, with $(\bar{\boldsymbol{A}}, \bar{\boldsymbol{B}}, \boldsymbol{C}) = (\boldsymbol{\Lambda}_t, \boldsymbol{K}_t^{\top}, \boldsymbol{Q}_t)$. Assuming ZOH discretization, the parameters for the corresponding continuous-time SSM defined in Eq. 11 can be expressed as follows:

$$\begin{cases} \bar{\boldsymbol{A}} = e^{\Delta \boldsymbol{A}} = \boldsymbol{\Lambda}_t, \\ \bar{\boldsymbol{B}} = (e^{\Delta \boldsymbol{A}} - \boldsymbol{I})\boldsymbol{A}^{-1}\boldsymbol{B} = \boldsymbol{K}_t^{\top}, \\ \boldsymbol{C} = \boldsymbol{Q}_t. \end{cases} \implies \begin{cases} \boldsymbol{A} = \frac{\ln(\boldsymbol{\Lambda}_t)}{\Delta}, \\ \boldsymbol{B} = \boldsymbol{A}(e^{\Delta \boldsymbol{A}} - \boldsymbol{I})^{-1}\boldsymbol{K}_t^{\top}, \\ \boldsymbol{C} = \boldsymbol{Q}_t \end{cases} \tag{14}$$

As a result, our recurrent SRA can be interpreted as a discretized ODE. Note that the ODE parameters $(\boldsymbol{A}, \boldsymbol{B}, \boldsymbol{C})$ in Eq. 14 are data-dependent with respect to the $t$-th observation $\boldsymbol{X}_t$. Therefore, this continuous-time ODE is actually a neural ODE, $\boldsymbol{S}'(t) = f(\boldsymbol{S}(t), t, \theta_t)$, with a differentiable neural network $f$ and data-dependent parameters $\theta_t = (\boldsymbol{A}, \boldsymbol{B}, \boldsymbol{C})$ (Chen et al., 2018). The continuous dynamics underlying the irregular sequences are models by a neural ODE as follows:

$$\boldsymbol{S}'(t) = \boldsymbol{A}\boldsymbol{S}(t) + \boldsymbol{B}\boldsymbol{X}(t) = f(\boldsymbol{S}(t), t, \theta), \; \boldsymbol{O}(t) = \boldsymbol{C}\boldsymbol{S}(t)$$

$$\text{where } \boldsymbol{A} = \frac{\ln(\boldsymbol{\Lambda}_t)}{\Delta}, \; \boldsymbol{B} = \boldsymbol{A}(e^{\Delta \boldsymbol{A}} - \boldsymbol{I})^{-1}\boldsymbol{K}_t^{\top}, \; \boldsymbol{C} = \boldsymbol{Q}_t, \; \boldsymbol{\Lambda}_t = \mathrm{diag}(\mathbf{1}\gamma_t). \tag{15}$$

Consequently, a single-head SRA serves as a discretized (neural) ODE model. When we generalize the multi-head scenario, TrajGPT can be considered as discretized ODEs, where each head of SRA corresponds to its own ODE and captures distinct dynamics.

## D  PROOF OF SSM DISCRETIZATION VIA ZOH RULE

To discretize the continuous-time model SSM, it has to compute the cumulative updates of the state $\boldsymbol{S}(t)$ over a discrete step size. For the continuous ODE in Eq. 11, we have a continuous-time integral as follows:

$$\boldsymbol{S}'(t) = \boldsymbol{A}\boldsymbol{S}(t) + \boldsymbol{B}\boldsymbol{X}(t)$$

$$\boldsymbol{S}(t+1) = \boldsymbol{S}(t) + \int_t^{t+1} (\boldsymbol{A}\boldsymbol{S}(\tau) + \boldsymbol{B}\boldsymbol{X}(\tau)) \, d\tau \tag{16}$$

In the discrete-time system, we need to rewrite the integral as we cannot obtain all values of $\boldsymbol{X}(\tau)$ over a continuous interval $t \to t+1$:

$$\boldsymbol{S}(t+1) = \boldsymbol{S}(t) + \sum_t^{t+1} (\boldsymbol{A}\boldsymbol{S}(\tau) + \boldsymbol{B}\boldsymbol{X}(\tau)\Delta\tau \tag{17}$$

We replace $\boldsymbol{X}(t)$ in the time derivative $\boldsymbol{S}'(t)$ as follows:

$$\boldsymbol{S}'(t) = \boldsymbol{A}\boldsymbol{S}(t) + \boldsymbol{B}\boldsymbol{X}(t)$$

$$\boldsymbol{S}'(t) - \boldsymbol{A}\boldsymbol{S}(t) = \boldsymbol{B}\boldsymbol{X}(t)$$

$$e^{-\boldsymbol{A}t}\boldsymbol{S}'(t) - e^{-\boldsymbol{A}t}\boldsymbol{A}\boldsymbol{S}(t) = e^{-\boldsymbol{A}t}\boldsymbol{B}\boldsymbol{X}(t)$$

$$\frac{d}{dt}\left(e^{-\boldsymbol{A}t}\boldsymbol{S}(t)\right) = e^{-\boldsymbol{A}t}\boldsymbol{B}\boldsymbol{X}(t)$$

$$e^{-\boldsymbol{A}t}\boldsymbol{S}(t) = \boldsymbol{S}(0) + \int_0^t e^{-\boldsymbol{A}\tau}\boldsymbol{B}\boldsymbol{X}(\tau)d\tau$$

$$\boldsymbol{S}(t) = e^{\boldsymbol{A}t}\boldsymbol{S}(0) + \int_0^t e^{\boldsymbol{A}(t-\tau)}\boldsymbol{B}\boldsymbol{X}(\tau)d\tau \tag{18}$$

By introducing a discrete step size $\Delta = t_{k+1} - t_k$, we transform the above equation to a discrete-time system as follows.

$$\boldsymbol{S}(t_{k+1}) = e^{\boldsymbol{A}(t_{k+1}-t_k)}\boldsymbol{S}(t_k) + \int_{t_k}^{t_{k+1}} e^{\boldsymbol{A}(t_{k+1}-\tau)}\boldsymbol{B}\boldsymbol{X}(\tau)d\tau$$

$$\boldsymbol{S}(t_{k+1}) = e^{\boldsymbol{A}(t_{k+1}-t_k)}\boldsymbol{S}(t_k) + \left(\int_{t_k}^{t_{k+1}} e^{\boldsymbol{A}(t_{k+1}-\tau)}d\tau\right)\boldsymbol{B}\boldsymbol{X}(t_k) \text{ (assuming } x(\tau) \approx x(t_k) \text{ over the interval)}$$

$$\boldsymbol{S}(t_{k+1}) = e^{\Delta\boldsymbol{A}}\boldsymbol{S}(t_k) + \boldsymbol{B}\boldsymbol{X}(t_k)\int_{t_k}^{t_{k+1}} e^{\boldsymbol{A}(t_{k+1}-\tau)}d\tau$$

$$\boldsymbol{S}(t_{k+1}) = e^{\Delta\boldsymbol{A}}\boldsymbol{S}(t_k) + \boldsymbol{B}\boldsymbol{X}(t_k)\int_0^{\Delta} e^{\boldsymbol{A}\tau'}d\tau' \text{ (letting } \tau' = t_{k+1} - \tau)$$

$$\boldsymbol{S}(t_{k+1}) = e^{\Delta\boldsymbol{A}}\boldsymbol{S}(t_k) + \boldsymbol{B}\boldsymbol{X}(t_k)\int_0^{\Delta} e^{\boldsymbol{A}\tau}d\tau$$

$$\boldsymbol{S}(t_{k+1}) = e^{\Delta\boldsymbol{A}}\boldsymbol{S}(t_k) + \boldsymbol{B}\boldsymbol{X}(t_k)\left(e^{\Delta\boldsymbol{A}} - \boldsymbol{I}\right)\boldsymbol{A}^{-1} \text{ (integral of matrix exponential function)}$$

$$\boldsymbol{S}_{k+1} = \boldsymbol{A}\boldsymbol{S}_k + \boldsymbol{B}\boldsymbol{X}_k \tag{19}$$

where the discretized state matrices $\bar{\boldsymbol{A}} = e^{\Delta\boldsymbol{A}}$ and $\bar{\boldsymbol{B}} = (e^{\Delta\boldsymbol{A}} - \boldsymbol{I})\boldsymbol{A}^{-1}\boldsymbol{B}$. Note that we apply the ZOH approach considering that $x(\tau)$ is constant between $t_k$ and $t_{k+1}$, we can rewrite the Eq. 19 by assuming $\boldsymbol{X}(\tau) \approx \boldsymbol{X}(t_k + 1)$:

$$\boldsymbol{S}(t_{k+1}) = e^{\boldsymbol{A}(t_{k+1}-t_k)}\boldsymbol{S}(t_k) + \int_{t_k}^{t_{k+1}} e^{\boldsymbol{A}(t_{k+1}-\tau)}\boldsymbol{B}\boldsymbol{X}(\tau)d\tau$$

$$\boldsymbol{S}(t_{k+1}) = e^{\boldsymbol{A}(t_{k+1}-t_k)}\boldsymbol{S}(t_k) + \left(\int_{t_k}^{t_{k+1}} e^{\boldsymbol{A}(t_{k+1}-\tau)}d\tau\right)\boldsymbol{B}\boldsymbol{X}(t_{k+1})$$

$$\boldsymbol{S}_{k+1} = \bar{\boldsymbol{A}}\boldsymbol{S}_k + \bar{\boldsymbol{B}}\boldsymbol{X}_{k+1} \tag{20}$$

The resulting equation is the discrete-time SSM using ZOH discretization in eq. 12.

**Derivation of $\bar{\boldsymbol{B}}$.** We use the equation $e^{\boldsymbol{A}\tau} = I + \boldsymbol{A}\tau + \frac{1}{2!}\boldsymbol{A}^2\tau^2 + \cdots$, we have this integral of exponential function of $\boldsymbol{A}$:

$$\bar{\boldsymbol{B}} = \int_0^{\Delta} e^{\boldsymbol{A}\tau}\boldsymbol{B}d\tau$$

$$= \int_0^{\Delta}\left(\boldsymbol{I} + \boldsymbol{A}\tau + \frac{1}{2!}\boldsymbol{A}^2\tau^2 + \cdots\right)d\tau\boldsymbol{B}$$

$$= \left(\boldsymbol{I}\Delta + \frac{1}{2}\boldsymbol{A}\Delta^2 + \frac{1}{3!}\boldsymbol{A}^2\Delta^3 + \cdots\right)\boldsymbol{B}$$

$$= \left(e^{\Delta\boldsymbol{A}} - \boldsymbol{I}\right)\boldsymbol{A}^{-1}\boldsymbol{B} \tag{21}$$

# E  DETAILS OF EXPERIMENTS

Table 5: Configurations of TrajGPT and other transformer baselines on the PopHR dataset. We set TrajGPT and all Transformer baseline to 7.5 million parameters.

| **TrajGPT** | |
| --- | --- |
| Decoder Layers | 8 |
| Heads | 4 |
| Dim ($Q$, $K$, $V$, FF) | 200,200,400,400 |
| **Transformer baselines including Encoder-decoder and Encoder-only models** | |
| Enc-Dec Layers | 4 & 4 |
| Encoder Layers | 8 |
| Decoder Layers | 8 |
| Heads | 4 |
| Dim ($Q$, $K$, $V$, FF) | 200,200,200,400 |

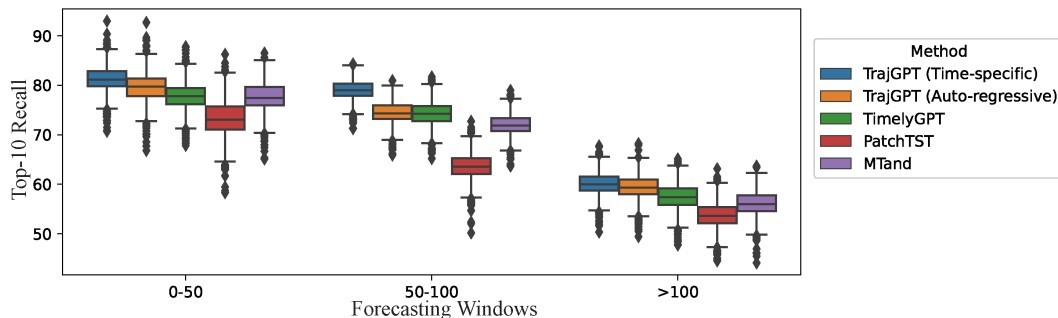

Figure 5: The distribution of top-10 recall performance for TrajGPT and baseline methods across three forecasting window sizes. The TrajGPT with time-specific inference achieves better and more stable performance compared with auto-regressive inference and other baselines.

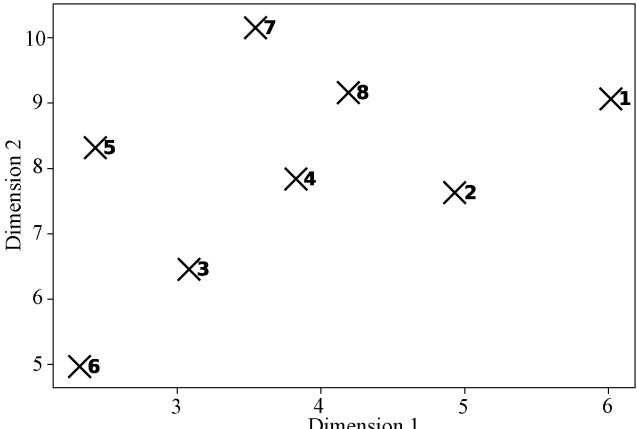

Figure 6: The distinct decay vectors $w_\gamma^{(h)}$ projected by UAMP, indicating that they capture different patterns.

