# OpenReview forum: "TrajGPT: Irregular Time-Series Representation Learning for Health Trajectory Analysis"
_ICLR.cc/2025/Conference — Submitted to ICLR 2025_

### Official Review · Reviewer_yGWz · 2024-10-30

**Soundness:** 4
**Presentation:** 4
**Contribution:** 4
**Rating:** 8
**Confidence:** 4

**Summary:**

This paper presents the Trajectory Generative Pre-trained Transformer (TrajGPT), a Transformer model designed to handle irregularly sampled time series, particularly within the healthcare domain. TrajGPT introduces a novel attention mechanism, Selective Recurrent Attention (SRA). Inspired by discretized ordinary differential equations (ODEs), the model achieves interpolation and extrapolation in irregular time series, excelling in trajectory prediction tasks. Experimental results show outstanding performance in zero-shot learning scenarios, underscoring TrajGPT’s capability to capture complex health dynamics from incomplete data.

**Strengths:**

- Comprehensive literature review.

- Thorough evaluation across different scenarios, with comparisons to multiple models and diverse datasets. The inclusion of interpretability through added figures is beneficial, and the ablation study is particularly valuable for assessing the contribution of each model component.

- The method is highly compelling, supported by a robust and thorough mathematical explanation.

- Novelty of the Attention Mechanism: The introduction of the Selective Recurrent Attention (SRA) module, which uses data-dependent decay, is an innovative contribution.

- Interpretation of TrajGPT as a Discretized ODE: Modeling TrajGPT as a discretized ODE is a valuable approach, allowing the model to capture continuous dynamics in irregular time series and perform time-specific inference, facilitating both interpolation and extrapolation in temporal data.

- Throughout the paper, I noted the similarities to State Space Models, which led me to find Appendix C particularly valuable.

**Weaknesses:**

There are few weaknesses in this paper:

- Since the relationship with State Space Models (SSMs) is discussed, it might have been interesting to include a method using Mamba to compare results with this type of model.

- Including a comparison with methods such as Multi-Gaussian Processes, which have been previously used for this type of approach, could strengthen the evaluation [1].

- Ultimately, as I understand it, this method has only been applied to univariate time series; it would be beneficial to explore its extension to multivariate time series in future work.

[1] Moor, M., Horn, M., Rieck, B., Roqueiro, D., & Borgwardt, K. (2019, October). Early recognition of sepsis with Gaussian process temporal convolutional networks and dynamic time warping. In Machine learning for healthcare conference (pp. 2-26). PMLR.

**Questions:**

- Do you believe this method could be easily extended to the context of multivariate time series?

- If expanded to multivariate time series, could it be combined with methods like GNNs to extract improved spatio-temporal representations?

- Despite the relationship, it remains unclear what the main advantages or potential inductive bias would be in using this method over Mamba blocks. Could you clarify this further? I would be happy to increase my score if the authors could expand on this information to clarify this important issue.

---

> ### Author Response · Authors · 2024-11-24
> **Response for Reviewer yGWz (Weakness 1,2,3, Question 1, 2)**
>
> Thanks for your careful review and constructive suggestions. Please find our response to your questions and concerns.
>
> **Response for both Weakness 1 and 2**:  In the revised version, we have included a discussion of the suggested MGP-TCN in Section 2.2. We also incorporated Mamba-1, Mamba-2, and MGP-TCN as baseline models in the PopHR experiments discussed in Section 5.2.
> We utilized the public GitHub codes for these three models with default setups. For the mamba models, we set the model parameters to around 7.5 million to ensure comparability with other Transformer baselines. These models employ next-token prediction pre-training and follow the same setup as TrajGPT for zero-shot learning, few-shot learning, and fine-tuning, applied to both forecasting and classification tasks. For MGP, originally designed for classification, we directly applied its code for insulin and CHF prediction. Since MGP-TCN does not support forecasting, we modified its classification output layer to generate predictions for all timesteps within the forecasting window.
>
> The table below, referenced from Table 1, compares the performance of TrajGPT with Mamba and MGP-TCN. The results demonstrate that TrajGPT with time-specific inference achieves the best performance across forecasting, insulin prediction, and CHF classification. Mamba-2 performs well on these two classification tasks with fine-tuning, while Mamba-1 and MGP-TCN fall behind.
>
> | Methods / Tasks (%)  | | **Forecast**   | | | **Insulin** | | | **CHF** | |
> |-|-|-|-|-|-|-|-|-|-|
> |   | K = 5    | K = 10   | K = 15   | S = 0   | S=5    | All   | S = 0   | S=5    | All   |
> | TrajGPT (Time-specific) | 57.4 (3.2) | 71.7 (2.6) | 84.1 (2.4) | 67.2 (3.1) | 70.2 (3.0) | 75.5 (2.6)  | 72.8 (2.4) | 75.9 (2.1) | 83.9 (2.0) |
> | TrajGPT (Auto-regressive) | 53.3 (3.9)  | 65.5 (3.4) | 77.2 (2.7)  | --- | --- | --- | --- | --- | --- |
> | Mamba-1 | 46.5 (3.6) | 62.4 (3.1) | 73.6 (2.6) | 61.5 (3.6) | 67.4 (3.2) | 72.5 (3.0) | 65.2 (3.1) | 70.1 (2.9) | 81.4 (2.4) |
> | Mamba-2 | 51.4 (3.2) | 69.8 (2.9) | 80.7 (2.5) | 64.6 (3.1) | 69.9 (2.8) | 74.8 (2.4) | 69.6 (2.7) | 73.9 (2.8) | 83.4 (2.3) |
> | MGP-TCN | 43.5 (3.5) | 57.2 (3.1) | 69.1 (2.9) | --- | --- | 73.9 (3.6) | --- | --- | 82.4 (3.5) |
>
> **R3 for Weakness 3**: To clarify, our model is designed to handle multivariate time series but in the context of discrete data. Specifically, we use a multinomial distribution for discrete tokens (e.g., one-hot-encoded diagnoses at any given time), where one token is 1 and all other tokens are 0 for each timestamp. In future work, we plan to extend the TrajGPT model to continuous multivariate time series (e.g., physiological signals such as ECG, clinical measurements such as MIMIC data, etc).
>
> **A1 for Question1**: As mentioned in R3, our model is designed to handle discrete multivariate time series. It is straightforward to extend it to continuous  multivariate time series by projecting the values from multiple variables into a token embedding. Hence, we only need to modify the input projection layer and keep the main architecture unchanged (e.g., [1, 2]).
>
> [1] TimelyGPT: Extrapolatable Transformer Pre-training for Long-term Time-Series Forecasting in Healthcare. ACM BCB 2024.
>
> [2] Time-LLM: Time series forecasting by reprogramming large language models. ICLR 2024
>
>
> **A2 for Question 2**: Thank you for this very interesting suggestion. In principal, TrajGPT can be combined with GNNs for spatio-temporal representations. Suppose we have a sequence of dynamic spatial graphs (i.e., one sequence per patient). Each patient- and time-specific graph may be obtained by processing the static biomedical knowledge graph via the patient's conditions and medications.
> At a given time point for a patient, the graph adjacency matrix can serve as a hard or soft constraint to compute the attention [3]. The resulting pooled embedding of the graph can then be fed to TrajGPT for temporal modeling.
>
> However, end-to-end training such model may be computationally challenging (e.g., tracking all graph- and time-attentions for backpropagation). Some existing work [4] employs a two-stage approach by first employing a pre-trained Transformer to encode sequence embeddings, which are then utilized to train a GNN, facilitating spatio-temporal representations.
>
> [3] Learning the graphical structure of electronic health records with graph convolutional transformer. AAAI 2020.
>
> [4] Pre-training enhanced spatial-temporal graph neural network for multivariate time series forecasting. KDD 2022.

---

> ### Author Response · Authors · 2024-11-24
> **Response for Reviewer yGWz (Question 3)**
>
> **A3 for Question3**: In this response, we clarify the advantages of TrajGPT over Mamba-1 and Mamba-2. While Mamba-1 is an SSM without linear attention, TrajGPT and Mamba-2 [5] are linear attention framework but with different decay gating mechanisms. TrajGPT offers a more expressive data-dependent decay mechanism than Mamba-2, which helps prevent catastrophic forgetting and maintains long-term dependencies. Additionally, TrajGPT surpasses Mamba-1 with efficient matrix representation for parallel forward-pass, as well as larger hidden dimensions and head size.
>
> The table below, adapted from Table 4 in [6], provides a comparison of these models:
>
> | **Model**        | **Recurrence**                                                 | **Output**                |
> |------------------|---------------------------------------------------------------|---------------------------|
> | Linear Attention |  $S_t = S_{t-1} + v_t k_t^\top$                             | $o_t = S_t q_t$     |
> | TrajGPT (ours)   | $S_t = \gamma_t S_{t-1} + v_t k_t^\top$                   | $o_t = S_t q_t$      |
> | Mamba            | $S_t = S_{t-1} \odot \exp(-(\alpha_t 1^\top) \odot \exp(A)) + (\alpha_t \odot v_t) k_t^\top $ | $o_t = S_t q_t + d \odot v_t $ |
> | Mamba-2          | $S_t = \beta_t S_{t-1} + v_t k_t^\top$                    | $o_t = S_t q_t$    |
>
> Specifically, Mamba-2 defines an SSM with both recurrent and parallel forms, called State Space Duality in original paper [5]. The recurrent form is:
>
> $$
>     h_t = A_t h_{t-1} + B_t X_t, Y_t = C_t^{\top} h_t \nonumber
> $$
>
> It also has a parallel form with a matrix representation:
>
> $$
>     Y = (L \circ C B^{\top} ) X
> $$
>
> where the matrix $L$ is defined as:
>
> \begin{equation}
>     L =
> \begin{bmatrix}
> 1 & &  & & & \\\\
> a_1 & 1 &  & & & \\\\
> a_2 a_1 & a_2 & 1  & & & \\\\
> \vdots & \vdots & \vdots & \ddots & & \\\\
> a_{T-1} \dots a_1 & a_{T-1} \dots a_2 & \dots & a_{T-1} & 1
> \end{bmatrix} \nonumber
> \end{equation}
>
>
> By mapping $(a_t, L, C, B, X) \xrightarrow{} (\beta_t, D, Q, K, V)$, Mamba-2 can be identified as a Linear Attention model with a data-dependent decay, similar to our TrajGPT. As discussed in our work, however, the computation of decay $\beta_t$ in Mamba-2 can cause numerical instability if the decay values are consistently small.  Mamba-2 addresses this by transitioning the computation to log space, but it still suffers from catastrophic forgetting if these  values remain consistently small. In contrast, our TrajGPT introduces a temporal term $\tau$ to control the range of $\gamma_t$,  defined as $\gamma_n = \text{Sigmoid} (X_n w_\gamma^\top)^{\frac{1}{\tau}}$. **Consequently, TrajGPT offers a more expressive decay mechanism than Mamba-2, effectively avoiding catastrophic forgetting and preserving long-term dependencies.**
>
> [5] Transformers are SSMs: Generalized Models and Efficient Algorithms Through Structured State Space Duality. ICML 2024.
>
> [6] Parallelizing Linear Transformers with the Delta Rule over Sequence Length. Neurips 2024.

---

> > ### Comment · Reviewer_yGWz · 2024-11-25
> > **Comprehensive Rebuttal and Substantial Improvements Elevate the Contribution of the Paper**
> >
> > The authors have responded to all my questions and have thoroughly updated the document where necessary. As a result, all my main concerns have been addressed, and I now have a better understanding of the contribution of this paper. Therefore, I am increasing my score from 6 to 8 and would like to commend the authors for their efforts during the review process.

---

> ### Author Response · Authors · 2024-11-25
> **Grateful for Reviewer's Feedback**
>
> Thank you for taking the time to review our responses and revised manuscript. We are grateful for your recognition of our efforts and the increased score.

---

### Official Review · Reviewer_MPfE · 2024-11-03

**Soundness:** 3
**Presentation:** 2
**Contribution:** 3
**Rating:** 5
**Confidence:** 4

**Summary:**

The paper introduces TrajGPT, a generative pre-trained Transformer designed for irregular time-series in healthcare applications. TrajGPT incorporates a novel linear Selective Recurrent Attention (SRA) mechanism that uses data-dependent decay to filter past information contextually. By interpreting the architecture as discretized ordinary differential equations (ODEs), the model effectively handles continuous dynamics, enabling it to interpolate and extrapolate for irregularly sampled data, thus enabling making direct predictions into desired future time step. TrajGPT is evaluated across trajectory forecasting, drug usage prediction, and phenotype classification, achieving strong zero-shot performance without task-specific tuning. Additionally, its design supports interpretability in health trajectory analysis, allowing clinicians to link disease trajectories with patient histories.

**Strengths:**

- Efficiency: TrajGPT achieves linear training and constant inference complexities, making it more scalable compared to traditional Transformers for long sequences.
- Performance and Generalizability: TrajGPT demonstrates impressive zero-shot capability across diverse tasks, highlighting its potential as a general-purpose tool for healthcare applications.
- Interpretability: The model’s architecture allows for meaningful health trajectory predictions and risk analysis, supporting clinical applications by enabling interpretative insights into disease progression.

**Weaknesses:**

- Ambiguity in the definition of ODEs: The proposed TrajGPT model has a form that the gradient of the trajectory at certain time is not only dependent on the state at that time, but also the input at that time. This data-dependent gradient no longer satisfies the definition of ODEs, but more towards the definition of PDEs. The authors should clarify this discrepancy and provide a more accurate interpretation of the model.
- Poorly justified visualization: In Fig 3a, the authors visualize the learned embeddings along with head specific decay vectors using UMAP. However, these decay vectors do not have same meaning as the embeddings, as one is intended to be a weight matrix of the input data while the other serves as a representation of the input data. Therefore, although these vectors share the same dimensionality, they should not be embedded in the same manifold space by using UMAP.
- Improvements in paper structure: The paper would benefit from a more structured presentation of the proposed model, for example, in section 3.1, the authors could avoid presenting the parallel formulation of the proposed model as it is not directly related to the main contributions of the paper. On the other hand, the experiment section could be expanded to provide more insights on the model's performance across different tasks, including more detailed explanation of the results and comparison with established medical findings.
- Absence of robustness analysis: As the paper proposes a novel model for healthcare applications, it is important to evaluate the robustness and potential biases of the model. For instance, as the model draws similarities to ODEs, it is crucial to investigate intrinsic biases in the model such as the limit cycle bahavior, which could lead to biased predictions in certain scenarios. The authors should provide a more comprehensive analysis on the robustness of the model to ensure its reliability in real-world applications.

**Questions:**

- Can you provide more insights on the discrepancy between the proposed model and the definition of ODEs? Since the model relies on the PDE form to make predictions in the future, the lack of future observation, i.e., inputs, will almost surely lead to biased or at least inaccurate predictions. Can you explain how the model handles this issue and how it can be improved?
- Can you justify for the visualization of the learned embeddings and decay vectors? Or seperate the visualization of the learned embeddings and decay vectors to avoid confusion. This will help readers to better understand the model's inner workings and the role of each component in the model and avoid misinterpretation of the results.
- Can you explain more on the results of the experiments and find more support from the literature to validate the model's performance? This will help to better understand the model's effectiveness and accuracy in real-world applications.

---

> ### Author Response · Authors · 2024-11-24
> **Response for Reviewer MPfE (Weakness 1,2,3)**
>
> We sincerely appreciate your thorough review and valuable feedback. We provide our responses to your insightful questions and concerns.
>
> **R1 for Weakness1**: First, we would like to clarify the definition of ODE and PDE. The primary distinction is that ODEs involve only one independent variable time $t$, while PDEs involve two or more independent variables, such as time $t$ and multiple time-independent variables $\mathbf{z} = [z_1, z_2, \ldots, z_D]$.
>
> According to [1], ODE is defined with the time $t$ and a time-dependent input $x(t)$, $\frac{d s}{dt}(t) = f(s(t), x(t))$, where the output $s$ is referred to as the ``state" in the SSM.  On the other hand, PDEs involve multiple independent variables in its definition $\partial_t s = f(s(t), \mathbf{z}, \partial_\mathbf{z} s, \partial_{\mathbf{z}\mathbf{z}} s, \ldots)$ [2], where $\mathbf{z}$ are time-independent variables. Same as ODEs, our model processes a time-dependent input $\mathbf{X}(t)$ and does not introduce additional time-independent variables.
>
> Furthermore, existing studies confirm that continuous-time SSMs are ODEs [3]. We have shown in Appendix C that our model is a discretization of a continuous-time SSM (i.e., an ODE). Therefore, our proposed SRA module is a discretized ODE.
>
> For a continuous-time SSM, we have $S^\prime (t) = A S(t) + B X(t)$, where $X(t)$ is time-dependent input and $S(t)$ is the state variable with $S_0 = 0$. This continuous-time SSM (i.e., ODE) can be discretized by various methods including Euler's method, zero-order hold (ZOH), and bilinear transform. In Appendix D, we show that our model is a discretization of SSM via ZOH.
>
> We revised our paper accordingly. In Section 3.2, we explicitly state that our SRA module involves only one independent variable $t$, making it a discretized ODE rather than a PDE. Additionally, we explicitly stated that Appendices C and D cover the theoretical connection to ODEs and the derivation of the ZOH discretization.
>
> [1] Parallelizing non-linear sequential models over the sequence length. ICLR 2024.
>
> [2] Message Passing Neural PDE Solvers. ICLR 2022.
>
> [3] Modeling sequences with structured state spaces. Dissertation of Albert Gu.
>
>
> **R2 for Weakness 2**: Although global head-specific decay vectors $\mathbf{w}^{(h)}_\gamma$ and token embeddings $\mathbf{x}_n$ are different types of information, we can still innovatively visualize them in the same manifold because of their dot product in the decay function : $\gamma^{(h)}_n = \text{Sigmoid} (\mathbf{x}_n \mathbf{w}^{(h)\top}_\gamma)$.
>
> Although global head-specific decay vectors $\mathbf{w}^{(h)}_\gamma$  and token embeddings $\mathbf{x}_n$ are different types of information, we can still innovatively visualize them in the same manifold because of their dot product in the decay function $\gamma^{(h)}_n = \text{Sigmoid} (\mathbf{x}_n \mathbf{w}^{(h)\top}_γ)$.
> Therefore, our UMAP visualization provides meaningful interpretation of the attention head by comparing the positions of their corresponding decay vectors with the token embeddings of known identities. For instance, when a category of token embeddings (e.g., endocrine/metabolic diseases) and a decay vector (e.g., head 8) are positioned closely in the latent space, this suggests a higher similarity between them.
> This proximity indicates that certain heads may specialize in certain disease categories, allowing the model to slow the forgetting processing and retain long-term dependencies.
>
> **R3 for Weakness 3**: The parallel form achieves linear training complexity, as discussed in Section 3.3.
> To improve clarity, however, we moved the derivation of the parallel form to the Appendix.
>
> To expand the experiment section, we conducted experiments on forecasting clinical events and early sepsis detection using the eICU dataset. In the updated version, we provided a discussion of eICU experimental results in Section 5.4 and Table 2, which show that TrajGPT achieves the highest performance in both forecasting tasks (top-10 and top-20 recall rates) and zero-shot sepsis classification.
>
> To address concerns about medical findings, we have updated Section 5.1 (Qualitative Analysis of Embeddings) and 5.3 (Trajectory Analysis) with additional references from the clinical domain. We have also expanded the discussion of our work's clinical impact in Section 5.3 (Trajectory Analysis) and Section 6 (Conclusion), incorporating relevant references.

---

> > ### Comment · Reviewer_MPfE · 2024-11-27
> >
> > Thank you for your response. However, it didn't answer my concerns.
> >
> > R1: I agree that the formulation of ODE is undebatable. However, in your proposed method, the "time-dependent input $x(t)$" is also in fact dependent on the patient or subject. Therefore, the system evolves with time w.r.t. that patient or subject. There is a major concern in the absence of this hidden variable in the method which naturally derives questions like the generalization ability to unseen patient.
> >
> > R2: I would kindly remind the authors that the UMAP works by keeping the structure of points unchanged in lower dimensional representation based on Euclidean distance. While I get your point trying to argue that Euclidean distance have relation to the dot product of their original vectors, this only holds for normalized vectors whose inner product to itself is 1, which is not enforced or demonstrated in your method. For a simple 2D example, chances can be that for group of points around $(1,1)$, a head-specific vector at $(1, .9)$ gets mapped to very close to these points, while another vector at $(2,2)$ isn't as close, but actually produce larger dot product with these points. Based on these, the UMAP **CANNOT** produce reliable interpretations of the results unless further ruling out these situations. This is not a critique to your proposed method, but rather a concern for the potentially misleading information generated from the results.
> >
> > As these major concerns are not addressed, I did not see the reason to change the current rating, so it will not be changed

---

> > > ### Author Response · Authors · 2024-11-28
> > > **Response to Review MPfE**
> > >
> > > Thank you for your insightful comments, which have been invaluable in helping us improve our work. To further address your feedback, we have outlined our responses below and made revisions accordingly.
> > >
> > > R1: Thanks for your feedback. To address your new question in your replay, we would like to clarify the generalization ability of the model with **pre-training**.
> > >
> > > Our model indeed learns time-dependent inputs that are subject-specific, but it does not face the generalization issue. Unlike traditional ODE-based models (e.g., neural ODE) trained from scratch, **TrajGPT leverages next-token prediction pre-training to learn generalizable temporal patterns [1], enabling effective adaptation to unseen subjects on the downstream tasks**.
> > >
> > > To demonstrate our model's generalizability, our study had evaluated zero-shot learning of our model on various downstream tasks. In the PopHR dataset, TrajGPT achieves the highest scores for zero shot learning in prediction tasks (K = 10, 15) and in the classification of diabetes and CHF, as well as the second best prediction results (K = 5). On the eICU dataset, TrajGPT achieves the highest zero-shot performance in forecasting (K=10, 20) and sepsis classification. In comparison, ODE-based models (e.g., ODE-RNN) do not have pre-training paradigm but only learn subject-specific inference, resulting in worse generalization capabilities, as noted by the reviewer.
> > >
> > > To further address your concern, we have revised the paper to include ablation studies on pre-training, as shown in the table below. These results demonstrate that pre-trained TrajGPT (with time-specific inference) achieves a 4.6\% improvement in zero-shot learning compared to the model trained from scratch. This highlights TrajGPT's ability to effectively generalize to unseen subjects and tasks in both forecasting and classification.
> > >
> > > | **Forecasting (K = 10)**       | **Time-specific inference** | **Auto-regressive inference** |
> > > |--------------------------------|----------------------------|-------------------------------|
> > > | TrajGPT (with Pre-training)    | 71.7    | 65.5    |
> > > | TrajGPT (without Pre-training) | 67.1      | 63.1   |
> > >
> > >
> > > R2: Thanks for your insightful and invaluable feedback. We recognize that the proximity in 2D space may not always imply higher similarity due to differences in magnitude, as illustrated by your provided example. In response, we have revised Fig.3 to focus solely on the token embeddings, removing the decay vectors $w^{(h)}_\gamma$. The updated figure still highlights the learned embeddings from distinct clusters corresponding to disease categories. Furthermore, we have projected the eight head decay vectors to distinct 2-D vectors using UMAP in Fig. 6, indicating that they capture different patterns.
> > > To address your concern regarding potentially misleading interpretations, we have revised the manuscript to exclude discussions linking token embeddings to decay vectors. These changes ensure that our revised version accurately represents the results.

---

> > > ### Author Response · Authors · 2024-12-02
> > > **Kindly Request for Rating Revision**
> > >
> > > We would like to express our gratitude to the reviewer for your valuable comments. We have carefullly considered all of your suggestions and made the corresponding updates to our submission. As today marks the final day of the rebuttal, we would like to kindly inquire if the reviewer would consider increasing the score.

---

> > > > ### Comment · Reviewer_MPfE · 2024-12-03
> > > >
> > > > Thank you for your additional clarification. They addressed most of my concerns, however, the theoretical basis of your argument that TrajGPT can generalize to unseen patients seems flawed, as the model does not own such capability in its mathematical nature. Moreover, the difference in the ablation study of pre-training is not significant enough to show that it gains generalization ability through pre-training, as there is no evidence or evaluation on the similarity between the pre-training PopHR dataset and downstream task datasets, so the increase in performance cannot simply attributed to single reasoning that it increases the generalization ability. The method and paper is good overall, although minor flaws like these cannot talk me into giving a better rating.

---

> > > > > ### Author Response · Authors · 2024-12-03
> > > > > **Response to Review MPfE**
> > > > >
> > > > > Thank you for your thoughtful feedback. We would like to clarify the generalization ability of our TrajGPT model in this final response.
> > > > >
> > > > > **Our TrajGPT model achieves generalization ability across unseen subjects through its next-token prediction pre-training task**. Previous studies (e.g., ELMo) have shown that pre-training can enhance model's generalization ability for various tasks [1]. GPT demonstrates that next-token prediction pre-training can generalize to unseen sequences [2]. Following this practice, our TrajGPT model uses next-token prediction pre-training to generalize to unseen subjects, which addresses the limitations of traditional ODE-based models without pre-training (e.g., ODE-RNN).
> > > > >
> > > > > While the mathematical nature of pre-training's generalization is an interesting topic, it is not the focus of this paper. We acknowledge that a deeper exploration of this topic would be valuable and should be addressed in future research. However, given the existing research progress in AI community, it does not have a concensus about the mathematical foundation of pre-training and its induced generalizabiltiy.
> > > > >
> > > > >
> > > > > [1] Deep contextualized word representations. NAACL, 2018.
> > > > >
> > > > > [2] Language Models are Unsupervised Multitask Learners. 2019.

---

> ### Author Response · Authors · 2024-11-24
> **Response for Reviewer MPfE (Weakness 4, Question 1,2,3)**
>
> **R4 for Weakness 4**: To address the concern about robustness, we have added a bootstrap evaluation of standard deviation in the revised manuscript. By resampling the dataset multiple times with replacement, we provide a robust estimate of the variance in our metrics. The table below provides a summary of TrajGPT's results, with the complete results available in the updated paper.
>
> | Methods / Tasks (%)  | | **Forecast**   | | | **Insulin** | | | **CHF** | |
> |-|-|-|-|-|-|-|-|-|-|
> |   | K = 5    | K = 10   | K = 15   | S = 0   | S=5    | All   | S = 0   | S=5    | All   |
> | TrajGPT (Time-specific) | 57.4 (3.2) | 71.7 (2.6) | 84.1 (2.4) | 67.2 (3.1) | 70.2 (3.0) | 75.5 (2.6)  | 72.8 (2.4) | 75.9 (2.1) | 83.9 (2.0) |
> | TrajGPT (Auto-regressive) | 53.3 (3.9)  | 65.5 (3.4) | 77.2 (2.7)  | --- | --- | --- | --- | --- | --- |
>
> The reviewer also raises an important point about intrinsic biases, such as the limit cycle commonly observed in ODEs. Our model addresses this by capturing trend patterns using an exponential decay mechanism, as discussed in existing studies [4, 5]. This mechanism enables efficient extrapolation by extending learned trend patterns to longer sequences beyond the training length. To demonstrate this, we include a risk trajectory analysis for diabetes and CHF (Fig. 4b and 4d). The results show that our model effectively captures clear trends of increasing risks with age, reflecting age-related vulnerability to chronic diseases. As a result, our model avoids the intrinsic biases of classical ODEs, such as limit cycles, which can lead to inaccurate predictions.
>
> [4] A Length-Extrapolatable Transformer. ACL 2023.
>
> [5] TimelyGPT: Extrapolatable Transformer Pre-training for Long-term Time-Series Forecasting in Healthcare. ACM BCB 2024.
>
> **A1 for Q1**: We have clarified that our model is a discretized ODE in our response R1. Similar to ODEs, our model makes predictions in a sequential manner. For each timestamp $t_{n}$, it computes the current state $S_{n}$ using the previous state $S_{n-1}$, and then use $S_{n}$ to predict the corresponding token $x_{n}$.
>
> However, the lack of future observations may cause biased prediction, leading to accumulated errors over time. TrajGPT mitigates this through time-specific inference, resulting in more stable and accurate predictions compared to other baselines, as discussed in Section 5.2 (Fig. 5). Specifically, it  predicts directly at any timestep using prior hidden states and current time, leveraging the additional information from the target timestamp. Therefore,  time-specific inference reduces computational steps and error accumulation.
>
> **A2 for Q2**: We have addressed the concerns about the UMAP visualization in our response R2.
>
> **A3 for Q3**: A3: We have addressed the concerns in our R3 by including additional discussion and relevant literature from the medical domain to support our result analysis.

---

### Official Review · Reviewer_5wW1 · 2024-11-03

**Soundness:** 3
**Presentation:** 3
**Contribution:** 3
**Rating:** 6
**Confidence:** 4

**Summary:**

The paper propose a novel transformer-based model tailored for irregularly-sampled time-series data, especially in healthcare domain.
TrajGPT features the Selective Recurrent Attention (SRA) mechanism, which incorporates a data-dependent decay to filter out irrelevant historical data, thus improving its adaptability to variable intervals in time-series data. The model is interpreted as discretized ODEs, enabling both interpolation and extrapolation, which is essential for accurate, time-specific predictions.

**Strengths:**

The authors provide clear motivations for their methods design. Especially,  the SRA mechanism design acts like a discretization of continuous-time ODE not only helping the model to learn the hidden continuous dynamics but also can achieve linear training complexity and constant inference complexity.
The visualizations in the result part looks really insightful and back up authors' assumption.

**Weaknesses:**

1. As the paper targeted representation-learning,  current representation-learning, like self-supervised learning for irregular time series lacks of discussion[1].
2. The paper only conducts experiments on PopHR dataset. Benchmark datasets such as MIMIC-III and physionet2012 are not included.

[1].Ranak Roy Chowdhury, Jiacheng Li, Xiyuan Zhang, Dezhi Hong, Rajesh K. Gupta, and Jingbo Shang. 2023. PrimeNet: pre-training for irregular multivariate time series. In Proceedings of the Thirty-Seventh AAAI Conference on Artificial Intelligence and Thirty-Fifth Conference on Innovative Applications of Artificial Intelligence and Thirteenth Symposium on Educational Advances in Artificial Intelligence (AAAI'23/IAAI'23/EAAI'23), Vol. 37. AAAI Press, Article 807, 7184–7192. https://doi.org/10.1609/aaai.v37i6.25876

**Questions:**

/

---

> ### Author Response · Authors · 2024-11-24
> **Response for Reviewer 5wW1**
>
> We are grateful for your constructive comments. Below we address each question and concern.
>
> **R1 for Weakness 1**: In the updated version, we have included a discussion of the suggested PrimeNet model in Section 2.1. We also incorporated it as a baseline in the experiments conducted on the PopHR and eICU datasets in Sections 5.2 and 5.4, respectively. We utilized the publicly available code from GitHub with default hyperparameters (https://github.com/ranakroychowdhury/PrimeNet). We follow the setups for pre-training, few-shot learning, and fine-tuning on the full dataset, while adopting the zero-shot learning from our work. Since PrimeNet does not support  forecasting, we modified its interpolation output layer to directly generate predictions for all timesteps within the forecasting window. For classification tasks, we used the original code without modifications.
>
> We present the results in the two tables below, corresponding to the experiments on the PopHR and eICU datasets, as referenced in Table 1 and Table 2 of the updated manuscript. TrajGPT with time-specific inference outperforms PrimeNet in forecasting tasks, attributable to its dynamic modeling and reduced error accumulation. Additionally, TrajGPT achieves superior performance in insulin usage prediction and early sepsis detection, while performing comparably to PrimeNet in CHF classification.
>
> | Methods / Tasks (%)  | | **Forecast**   | | | **Insulin** | | | **CHF** | |
> |-|-|-|-|-|-|-|-|-|-|
> |   | K = 5    | K = 10   | K = 15   | S = 0   | S=5    | All   | S = 0   | S=5    | All   |
> | TrajGPT (Time-specific) | 57.4 (3.2) | 71.7 (2.6) | 84.1 (2.4) | 67.2 (3.1) | 70.2 (3.0) | 75.5 (2.6)  | 72.8 (2.4) | 75.9 (2.1) | 83.9 (2.0) |
> | TrajGPT (Auto-regressive) | 53.3 (3.9)  | 65.5 (3.4) | 77.2 (2.7)  | --- | --- | --- | --- | --- | --- |
> | PrimeNet | 52.5 (3.2) | 69.7 (2.8) | 81.8 (2.3) | 65.6 (3.0) | 69.5 (2.9) | 73.8 (2.7) | 71.5 (2.7) | 75.5 (2.9) | 84.0 (2.4) |
>
> | Methods / Tasks (%) | **Forecast** | | **Sepsis** | |
> |-|-|-|-|-|
> | | K = 10 | K = 20 | S = 0 | All |
> | TrajGPT (Time-specific) | 57.8 (2.9) | 69.3 (2.1) | 45.1 (2.7) | 51.3 (2.4) |
> | TrajGPT (Auto-regressive) | 54.1 (3.2) | 64.9 (2.3)  | --- | --- |
> | PrimeNet | 53.4 (2.3) | 67.5 (2.0) | 44.0 (2.3) | 50.6 (1.9) |
>
> **R2 for Weakness 2**: Thanks you for your feedback regarding the evaluation and benchmarking. We experimented with the publicly available eICU dataset from PhysioNet because it provides irregularly-sampled time series with discrete diagnoses and drugs provided to patients during their ICU stay. We extracted a dataset containing 288 unique PheCodes and 228 unique drugs. The final dataset includes 139,367 patients, with each patient having an average of 19 drugs and 3 ICD codes recorded within 15-minute intervals. We conducted the forecasting tasks on irregular clinical events (diagnoses and drugs) and early detection of sepsis.
>
> We have updated the paper to include a description of dataset and preprocessing in Section 4.1, as well as an explanation of the experimental design in Section 4.3. Additionally, the updated Section 5.4 includes a discussion of the eICU experimental results, with the outcomes presented in Table 2. Below, we show the table on eICU benchmarking, which evaluate TrajGPT and baselines on the top-K diagnosis/medication forecasting and sepsis prediction. The experimental results highlight that TrajGPT achieves the highest performance in both forecasting tasks (top-10 and top-20 recall rates) and zero-shot sepsis classification.
>
>
>
> | Methods / Tasks (%)  | **Forecasting** | | **Sepsis**  | |
> |-|-|-|-|-|
> | | K = 10 | K = 20 | S = 0 | All |
> | TrajGPT (Time-specific)| **57.8 (2.9)** | **69.3 (2.1)** | **45.1 (2.7)** | 51.3 (2.4) |
> | TrajGPT (Auto-regressive)| 54.1 (3.2) | 64.9 (2.3) | --- | --- |
> | TimelyGPT| 56.9 (3.2) | 67.1 (2.4) | 42.0 (2.5) | 48.5 (2.2) |
> | PatchTST | 55.2 (2.7) | 66.0 (1.7) | 44.5 (2.2) | 51.8 (1.8) |
> | TimesNet | 52.9 (3.1) | 60.3 (2.3) | 41.2 (3.1) | 47.5 (2.6) |
> | ContiFormer | 57.1 (2.2) | 66.8 (2.2) | 41.7 (2.5) | 50.6 (2.8) |
> | PrimeNet | 53.4 (2.3) | 67.5 (2.0) | 44.0 (2.3) | 51.2 (1.9) |
> | Mamba-2 | 53.7 (2.8) | 63.2 (2.3) | 42.6 (2.8) | 48.9 (2.3) |
> | MTand | 53.9 (2.4) | 67.4 (1.6)  | --- | **52.5 (2.1)** |
> | ODE-RNN | 55.7 (3.4) | 67.8 (2.8) | --- | 49.2 (2.9) |

---

> > ### Comment · Reviewer_5wW1 · 2024-11-25
> >
> > Thank you for providing more details and experiments results, I will keep the score.

---

> > > ### Author Response · Authors · 2024-11-25
> > > **Grateful for Reviewer's Feedback**
> > >
> > > We sincerely appreciate your constructive comments and feedbacks, as they were invaluable in helping us enhance the clarity and contribution of our work.

---

### Official Review · Reviewer_i6Sx · 2024-11-05

**Soundness:** 3
**Presentation:** 3
**Contribution:** 3
**Rating:** 6
**Confidence:** 5

**Summary:**

The paper proposes an approach for representation learning of irregularly sampled time-series data called Trajectory Generative Pre-trained Transformer (TrajGPT). The TrajGPT architecture employs a Selective Recurrent Attention (SRA) mechanism that allows it to selectively focus on relevant past information and ignore irrelevant data. The proposed approach is able to capture continuous dynamics and perform interpolation and extrapolation. The proposed approach achieves strong performance on several tasks, including forecasting disease trajectories, predicting drug usage, and classifying patient phenotypes. It also show good zero-shot performance on several tasks and provides interpretable results.

**Strengths:**

- The paper focuses on an important problem of learning from irregularly sampled time series which is important for several practical use cases.
- The paper introduces a novel architecture for modeling irregularly sampled time series- selective recurrent attention (SRA). SRA enables time-specific inference that leads to better imputation and forecasting.
- Experiments show that the framework achieves strong zero-shot performance across multiple tasks and achieves comparable results with SOTA in the finetuning setup.
- The paper is well written and easy to follow.

**Weaknesses:**

- The evaluation of TrajGPT is limited, as it does not include benchmark irregularly sampled time series datasets such as PhysioNet and MIMIC-III. This makes it difficult to compare the performance of TrajGPT with other state-of-the-art models in this space despite the authors doing a good job of comparing on PopHR dataset.
- The paper overlooks important relevant work [1] and fails to provide a comparison with [1] which is the state-of-the-art for imputation in irregularly sampled time series.
- The paper lacks insight into why time-specific inference leads to better results than auto-regressive inference, which is the method used to train the model.


### References
[1] Heteroscedastic Temporal Variational Autoencoder For Irregularly Sampled Time Series, International Conference on Learning Representations, 2022.

**Questions:**

Listed them out in the weaknesses section.

---

> ### Author Response · Authors · 2024-11-24
> **Response for Reviewer i6Sx**
>
> We appreciate the reviewer’s comments. We update our paper and address your concerns here:
>
>
> **R1 for Weakness 1**: Thanks you for your feedback regarding the evaluation and benchmarking. We experimented with the publicly available eICU dataset from PhysioNet because it provides irregularly-sampled time series with discrete diagnoses and drugs provided to patients during their ICU stay. We extracted a dataset containing 288 unique PheCodes and 228 unique drugs. The final dataset includes 139,367 patients, with each patient having an average of 19 drugs and 3 ICD codes recorded within 15-minute intervals. We conducted the forecasting tasks on irregular clinical events (diagnoses and drugs) and early detection of sepsis.
>
> We have updated the paper to include a description of dataset and preprocessing in Section 4.1, as well as an explanation of the experimental design in Section 4.3. Additionally, the updated Section 5.4 includes a discussion of the eICU experimental results, with the outcomes presented in Table 2. Below, we show the table on eICU benchmarking, which evaluate TrajGPT and baselines on the top-K diagnosis/medication forecasting and sepsis prediction. The experimental results highlight that TrajGPT achieves the highest performance in both forecasting tasks (top-10 and top-20 recall rates) and zero-shot sepsis classification.
>
> | Methods / Tasks (%)  | **Forecasting** | | **Sepsis**  | |
> |-|-|-|-|-|
> | | K = 10 | K = 20 | S = 0 | All |
> | TrajGPT (Time-specific)| **57.8 (2.9)** | **69.3 (2.1)** | **45.1 (2.7)** | 51.3 (2.4) |
> | TrajGPT (Auto-regressive)| 54.1 (3.2) | 64.9 (2.3) | --- | --- |
> | TimelyGPT| 56.9 (3.2) | 67.1 (2.4) | 42.0 (2.5) | 48.5 (2.2) |
> | PatchTST | 55.2 (2.7) | 66.0 (1.7) | 44.5 (2.2) | 51.8 (1.8) |
> | TimesNet | 52.9 (3.1) | 60.3 (2.3) | 41.2 (3.1) | 47.5 (2.6) |
> | ContiFormer | 57.1 (2.2) | 66.8 (2.2) | 41.7 (2.5) | 50.6 (2.8) |
> | PrimeNet | 53.4 (2.3) | 67.5 (2.0) | 44.0 (2.3) | 51.2 (1.9) |
> | Mamba-2 | 53.7 (2.8) | 63.2 (2.3) | 42.6 (2.8) | 48.9 (2.3) |
> | MTand | 53.9 (2.4) | 67.4 (1.6)  | --- | **52.5 (2.1)** |
> | ODE-RNN | 55.7 (3.4) | 67.8 (2.8) | --- | 49.2 (2.9) |
>
> **R2 for Weakness 2**:  In the updated version, we have included a discussion of the suggested HeTVAE in Section 2.2. We also incorporated it as a baseline in the PopHR experiments discussed in Section 5.2. We utilized the publicly available code from GitHub with default hyperparameters (https://github.com/reml-lab/hetvae). While HeTVAE was originally designed for interpolation tasks, we modified its output linear layer to accommodate our forecasting and classification tasks. For forecasting, the output layer directly generates predictions for all timesteps within the forecasting window. For classification, it directly outputs the predicted class. Since HeTVAE does not support a representation learning approach, we trained it from scratch on the entire dataset for the insulin and CHF classification tasks.
>
> The table below, referenced from Table 1, compares the performance of TrajGPT and HeTVAE.  While HeTVAE demonstrates strong forecasting performance for top-K recall with K=10 and K=15, TrajGPT with time-specific inference surpasses it by 1.6\% and 0.9\%, respectively. However, HeTVAE, originally designed for generative interpolation tasks, performs poorly on discriminative prediction tasks such as insulin usage and CHF phenotyping.
>
> | Methods / Tasks (%)  | | **Forecast**   | | | **Insulin** | | | **CHF** | |
> |-|-|-|-|-|-|-|-|-|-|
> |   | K = 5    | K = 10   | K = 15   | S = 0   | S=5    | All   | S = 0   | S=5    | All   |
> | TrajGPT (Time-specific) | 57.4 (3.2) | 71.7 (2.6) | 84.1 (2.4) | 67.2 (3.1) | 70.2 (3.0) | 75.5 (2.6) | 72.8 (2.4) | 75.9 (2.1) | 83.9 (2.0)  |
> | TrajGPT (Auto-regressive) | 53.3 (3.9) | 65.5 (3.4) | 77.2 (2.7) | --- | --- | --- | --- | ---  | ---  |
> | HeTVAE | 51.1 (3.9) | 70.1 (3.4) | 83.2 (3.2) | ---     | ---  | 71.4 (3.6) | ---     | ---  | 81.6 (3.2) |
>
> **R3 for Weakness 3**: Thank you for your question. We have revised Section 3.2 to clarify this. For auto-regressive method, it generates samples sequentially and selects from irregularly spaced timestamps, which can lead to error accumulation. Our proposed time-specific inference directly extrapolates to the target timestep using prior hidden states and the gap from the last observed  timestep. Therefore, it reduces computational steps and error accumulation for better performance.

---

### Meta-Review · Area_Chair_HxTi · 2024-12-21

**Metareview:**

**Summary**: The paper proposes a GPT framework for modeling irregularly sampled sequential data incorporating a selective attention mechanism to enable the model to adaptively use historical context. Empirical evaluation demonstrates good zero-short performance on healthcare data and under missingness.

**Strengths**:
1. Comparison to baselines and prior work is comprehensive.
2. Empirical evaluation includes ablations of specific benefits of the model (with and without pre-training)
3. Reviewers have found the model to be fairly effective at modeling patient trajectories

**Weaknesses**:
1. Initial version of the paper did not have benchmark evaluations with respect to MIMIC-IV and eICU data. Subsequent rebuttal has incorporated results for these standard benchmark datasets.
2. There are concerns raised about the framing and analogy to ODEs. I agree with some of these concerns and over-interpretation.
3. Initial version of the paper also did not benchmark against State-space model analogues like MAMBA and did not have uncertainty quantification. Which have since been provided as a response in the rebuttal.
4. Over-interpretation and issues linger in the UMAP visualization of patient trajectories.

**Recommendation**:
Significant concerns about the paper have been addressed in the rebuttal, the updated version of the paper is improved in presentation. However, some fundamental technical framing issues remain. First, I do agree that the analogy to ODEs is slightly problematic and should be significantly tempered. Second, I believe all additional empirical evaluation provided in author response should be incorporated rigorously in the final version with reasonable bootstrapping. Finally, ablations should be more systematic. Considering all these issues, I believe the paper will benefit from an additional round of peer review and therefore I am recommending a reject.

**Additional Comments On Reviewer Discussion:**

No additional raised during discussion.

---

### Decision · Program_Chairs · 2025-01-22

Reject